# Deep Variational Multivariate Information Bottleneck - A Framework for Variational Losses

## Abstract

Variational dimensionality reduction methods are known for their high accuracy, generative abilities, and robustness. They have many theoretical justifications. Here we introduce a framework to unify existing variational methods and design new ones. The framework is based on an interpretation of the multivariate information bottleneck, in which two Bayesian networks are traded off against one another. The first network is a compression or encoder graph, which specifies what information to keep when compressing the data. The second network is a decoder graph, which specifies a generative model for the data. Using this framework, we rederive existing dimensionality reduction methods such as the deep variational information bottleneck (DVIB), beta variational auto-encoders (beta-VAE), deep variational canonical correlation analysis (DVCCA), etc. The framework naturally introduces a trade-off parameter between compression and reconstruction in the DVCCA family of algorithms, resulting in the new beta-DVCCA family. In addition, we derive a new variational dimensionality reduction method, deep variational symmetric informational bottleneck (DVSIB), which simultaneously compresses two variables to preserve information between their compressed representations. We implement all of these algorithms and evaluate their ability to produce shared low dimensional latent spaces on benchmark datasets. We show that algorithms that are better matched to the structure of the data (beta-DVCCA and DVSIB in our case) produce better latent spaces as measured by classification accuracy and the dimensionality of the latent variables. We believe that this framework can be used to unify other multi-view representation learning algorithms and to derive and implement novel problem-specific loss functions.

## 1 Introduction

Large dimensional multi-modal datasets are abundant in multimedia systems utilized for language modeling (Xu et al., 2016; Zhou et al., 2018; Rohrbach et al., 2017; Sanabria et al., 2018; Goyal et al., 2017; Wang et al., 2018; Hendrycks et al., 2020), neural control of behavior studies (Steinmetz et al., 2021; Urai et al., 2022; Krakauer et al., 2017; Pang et al., 2016), multi-omics approaches in systems biology (Clark et al., 2013; Zheng et al., 2017; Svensson et al., 2018; Huntley et al., 2015; Lorenzi et al., 2018), and many other domains. Such data come with the curse of dimensionality, making it hard to learn the relevant statistical correlations from samples. The problem is made even harder by the data often containing information that is irrelevant to the specific questions one asks. To tackle these challenges, a myriad of dimensionality reduction (DR) methods have emerged. By preserving certain aspects of the data while discarding the remainder, DR can decrease the complexity of the problem, yield clearer insights, and provide a foundation for more refined modeling approaches.

DR techniques span linear methods like Principal Component Analysis (PCA) (Hotelling, 1933), Partial Least Squares (PLS) (Wold et al., 2001), Canonical Correlations Analysis (CCA) (Hotelling, 1936), and regularized CCA (Vinod, 1976; Årup Nielsen et al., 1998), as well as nonlinear approaches, including Autoencoders (AE) (Hinton & Salakhutdinov, 2006), Deep CCA (Andrew et al., 2013), Deep Canonical Correlated AE (Wang et al., 2015), Correlational Neural Networks (Chandar et al., 2016), Deep Generalized CCA (Benton et al., 2017), and Deep Tensor CCA (Wong et al., 2021). Of particular interest to us are variational methods, such as Variational Autoencoders (VAE) (Kingma

& Welling, 2014), beta-VAE (Higgins et al., 2016), Joint Multimodal VAE (JMVAE) (Suzuki et al., 2016), Deep Variational CCA (DVCCA) (Wang et al., 2016), Deep Variational Information Bottleneck (DVIB) (Alemi et al., 2017), Variational Mixture-of-experts AE (Shi et al., 2019), and Multiview Information Bottleneck (Federici et al., 2020b). These DR methods use deep neural networks and variational approximations to learn robust and accurate representations of the data, while, at the same time, often serving as generative models for creating samples from the learned distributions.

There are many theoretical derivations and justifications for variational DR methods (Kingma & Welling, 2014; Higgins et al., 2016; Suzuki et al., 2016; Wang et al., 2016; Karami & Schuurmans, 2021; Qiu et al., 2022; Alemi et al., 2017; Bao, 2021; Lee & Van der Schaar, 2021; Wang et al., 2019; Wan et al., 2021; Federici et al., 2020a; Huang et al., 2022; Hu et al., 2020). This diversity of derivations, while enabling adaptability, often leaves researchers with no principled ways for choosing a method for a particular application, for designing new methods with distinct assumptions, or for comparing methods to each other.

Here, we introduce the Deep Variational Multivariate Information Bottleneck (DVMIB) framework, offering a unified mathematical foundation for many variational DR methods. Our framework is grounded in the multivariate information bottleneck loss function (Tishby et al., 2000; Friedman et al., 2013). This loss, amenable to approximation through upper and lower variational bounds, provides a system for implementing diverse DR variants using deep neural networks. We demonstrate the framework's efficacy by deriving the loss functions of many existing variational DR methods starting from the same principles. Furthermore, our framework naturally allows the adjustment of trade-off parameters, leading to generalizations of these existing methods. For instance, we generalize DVCCA to $\beta$-DVCCA. The framework further allows us to introduce and implement in software novel DR methods. We view the DVMIB framework, with its uniform information bottleneck language, conceptual clarity of translating statistical dependencies in data via graphical models of encoder and decoder structures into variational losses, the ability to unify existing approaches, and easy adaptability to new scenarios as one of the two main contributions of our work.

Beyond its unifying role, our framework offers a principled approach for deriving problem-specific loss functions using domain-specific knowledge. Thus, we anticipate its application for multi-view representation learning across diverse fields. To illustrate this, we use the framework to derive a novel dimensionality reduction method, the Deep Variational Symmetric Information Bottleneck (DVSIB), which compresses two random variables into two distinct latent variables that are maximally informative about one another. This new method produces better representations of classic datasets than previous approaches. Introduction of DVSIB is the second major contribution of our paper.

The paper is structured as follows. First, we introduce the underlying mathematics and the implementation of the DVMIB framework. We then explain how to use it to generate new DR methods. In Tbl. 1, we present several known and newly derived variational methods, illustrating how easily they can be derived within the framework. As a proof of concept, we then benchmark *simple* computational implementations of methods in Tbl. 1 against the Noisy MNIST dataset. Appendices present detailed treatment of all terms in variational losses in our framework, discussion of multi-view generalizations, empirical tests of DVSIB on an additional dataset (MIR-Flickr)(Huiskes & Lew, 2008), and more details —including visualizations— of performance of many methods on the Noisy MNIST.

## 2 MULTIVARIATE INFORMATION BOTTLENECK FRAMEWORK

We represent DR problems similar to the Multivariate Information Bottleneck (MIB) of Friedman et al. (2013), which is a generalization of the more traditional Information Bottleneck algorithm (Tishby et al., 2000) to multiple variables. The reduced representation, is achieved as a trade-off between two Bayesian networks. Bayesian networks are directed acyclic graphs that provide a factorization of the joint probability distribution, $P(X_1, X_2, X_3, .., X_N) = \prod_{i=1}^{N} P(X_i | Pa_{X_i}^G)$, where $Pa_{X_i}^G$ is the set of parents of $X_i$ in graph $G$. The multiinformation (Studený & Vejnarová, 1998) of a Bayesian network is defined as the Kullback-Leibler divergence between the joint probability distribution and the product of the marginals, and it serves as a measure of the total correlations among the variables, $I(X_1, X_2, X_3, ..., X_N) = D_{KL}(P(X_1, X_2, X_3, ..., X_N) \| P(X_1)P(X_2)P(X_3)...P(X_N))$. For a Bayesian network, the multiinformation reduces to the sum of all the local informations $I(X_1, X_2, ..X_N) = \sum_{i=1}^{N} I(X_i; Pa_{X_i}^G)$ (Friedman et al., 2013).

The first of the Bayesian networks is an encoder (compression) graph, which models how compressed (reduced, latent) variables are obtained from the observations. The second network is a decoder graph, which specifies a generative model for the data from the compressed variables, i.e., it is an alternate factorization of the distribution. In MIB, the information of the encoder graph is minimized, ensuring strong compression (corresponding to the approximate posterior). The information of the decoder graph is maximized, promoting the most accurate model of the data (corresponding to maximizing the log-likelihood). As in IB (Tishby et al., 2000), the trade-off between the compression and reconstruction is controlled by a trade-off parameter $\beta$:

$$L = I_{\text{encoder}} - \beta I_{\text{decoder}}. \tag{1}$$

In this work, our key contribution is in writing an explicit variational loss for typical information terms found in both the encoder and the decoder graphs. All terms in the decoder graph use samples of the compressed variables as determined from the encoder graph. If there are two terms that correspond to the same information in Eq. (1), one from each of the graphs, they do not cancel each other since they correspond to two different variational expressions. For pedagogical clarity, we do this by first analyzing the Symmetric Information Bottleneck (SIB), a *special case* of MIB. We derive the bounds for three types of information terms in SIB, which we then use as building blocks for all other variational MIB methods in subsequent Sections.

## 2.1 Deep Variational Symmetric Information Bottleneck

The Deep Variational Symmetric Information Bottleneck (DVSIB) simultaneously reduces a pair of datasets $X$ and $Y$ into two separate lower dimensional compressed versions $Z_X$ and $Z_Y$. These compressions are done at the same time to ensure that the latent spaces are maximally informative about each other. The joint compression is known to decrease data set size requirements compared to individual ones (Martini & Nemenman, 2023). Having distinct latent spaces for each modality usually helps with the interpretability. For example, $X$ could be the neural activity of thousands of neurons, and $Y$ could be the recordings of joint angles of the animal. Rather than one latent space representing both, separate latent spaces for the neural activity and the joint angles are sought. By maximizing compression as well as $I(Z_X, Z_Y)$, one constructs the latent spaces that capture only the neural activity pertinent to joint movement and only the movement that is correlated with the neural activity (cf. Pang et al. (2016)). Many other applications could benefit from a similar DR approach.

In Fig. 1, we define two Bayesian networks for DVSIB, $G_{\text{encoder}}$ and $G_{\text{decoder}}$. $G_{\text{encoder}}$ encodes the compression of $X$ to $Z_X$ and $Y$ to $Z_Y$. It corresponds to the factorization $p(x, y, z_x, z_y) = p(x, y)p(z_x|x)p(z_y|y)$ and the resultant $I_{\text{encoder}} = I^E(X; Y) + I^E(X; Z_X) + I^E(Y; Z_Y)$. The $I^E(X, Y)$ term does not depend on the compressed variables, does not affect the optimization problem, and hence is discarded in what follows. $G_{\text{decoder}}$ represents a generative model for $X$ and $Y$ given the compressed latent variables $Z_X$ and $Z_Y$. It corresponds to the factorization $p(x, y, z_x, z_y) = p(z_x)p(z_y|z_x)p(x|z_x)p(y|z_y)$ and the resultant $I_{\text{decoder}} = I^D(Z_X; Z_Y) + I^D(X; Z_X) +$

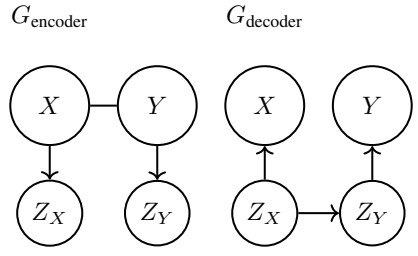

Figure 1: The encoder and decoder graphs for DVSIB.

$I^D(Y; Z_Y)$. Combing the informations from both graphs and using Eq. (1), we find the SIB loss:

$$L_{\text{SIB}} = I^E(X; Z_X) + I^E(Y; Z_Y) - \beta \left( I^D(Z_X; Z_Y) + I^D(X; Z_X) + I^D(Y; Z_Y) \right). \tag{2}$$

Note that information in the encoder terms is minimized, and information in the decoder terms is maximized. Thus, while it is tempting to simplify Eq. (2) by canceling $I^E(X; Z_X)$ and $I^D(X; Z_X)$, this would be a mistake. Indeed, these terms come from different factorizations: the encoder corresponds to learning $p(z_x|x)$, and the decoder to $p(x|z_x)$.

While the DVSIB loss may appear similar to previous models, such as MultiView Information Bottleneck (MVIB) (Federici et al., 2020b) and Barlow Twins (Zbontar et al., 2021), it is conceptually distinct. For example, MVIB aims to generate latent variables that are as similar to each other as possible, sharing the same domain. DVSIB, however, endeavors to produce distinct latent representations, which could potentially have different units or dimensions, while maximizing mutual

information between them. Barlow Twins architecture on the other hand appears to have two latent subspaces while in fact they are one latent subspace that is being optimized by a regular information bottleneck.

We now follow a procedure and notation similar to Alemi et al. (2017) and construct variational bounds on all $I^E$ and $I^D$ terms. Terms without leaf nodes, i. e., $I^D(Z_X, Z_Y)$, require new approaches.

## 2.2 VARIATIONAL BOUNDS ON DVSIB ENCODER TERMS

The information $I^E(Z_X; X)$ corresponds to compressing the random variable $X$ to $Z_X$. Since this is an encoder term, it needs to be minimized in Eq. (2). Thus, we seek a variational bound $I^E(Z_X; X) \leq \tilde{I}^E(Z_X; X)$, where $\tilde{I}^E$ is the variational version of $I^E$, which can be implemented using a deep neural network. We find $\tilde{I}^E$ by using the positivity of the Kullback–Leibler divergence. We make $r(z_x)$ be a variational approximation to $p(z_x)$. Then $D_{\mathrm{KL}}(p(z_x)\|r(z_x)) \geq 0$, so that $-\int dz_x p(z_x) \ln(p(z_x)) \leq -\int dz_x p(z_x) \ln(r(z_x))$. Thus, $-\int dx dz_x p(z_x, x) \ln(p(z_x)) \leq -\int dx dz_x p(z_x, x) \ln(r(z_x))$. We then add $\int dx dz_x p(z_x, x) \ln(p(z_x|x))$ to both sides and find:

$$I^E(Z_X; X) = \int dx dz_x p(z_x, x) \ln\left(\frac{p(z_x|x)}{p(z_x)}\right) \leq \int dx dz_x p(z_x, x) \ln\left(\frac{p(z_x|x)}{r(z_x)}\right) \equiv \tilde{I}^E(Z_X; X). \tag{3}$$

We further simplify the variational loss by approximating $p(x) \approx \frac{1}{N} \sum_{i=1}^N \delta(x - x_i)$, so that:

$$\tilde{I}^E(Z_X; X) \approx \frac{1}{N} \sum_{i=1}^N \int dz_x p(z_x|x_i) \ln\left(\frac{p(z_x|x_i)}{r(z_x)}\right) = \frac{1}{N} \sum_{i=1}^N D_{\mathrm{KL}}(p(z_x|x_i)\|r(z_x)). \tag{4}$$

The term $I^E(Y; Z_Y)$ can be treated in an analogous manner, resulting in:

$$\tilde{I}^E(Z_Y; Y) \approx \frac{1}{N} \sum_{i=1}^N D_{KL}(p(z_y|y_i)\|r(z_y)). \tag{5}$$

## 2.3 VARIATIONAL BOUNDS ON DVSIB DECODER TERMS

The term $I^D(X; Z)$ corresponds to a decoder of $X$ from the compressed variable $Z_X$. It is maximized in Eq. (2). Thus, we seek its variational version $\tilde{I}^D$, such that $I^D \geq \tilde{I}^D$. Here, $q(x|z_x)$ will serve as a variational approximation to $p(x|z_x)$. We use the positivity of the Kullback-Leibler divergence, $D_{\mathrm{KL}}(p(x|z_x)\|q(x|z_x)) \geq 0$, to find $\int dx\, p(x|z_x) \ln(p(x|z_x)) \geq \int dx\, p(x|z_x) \ln(q(x|z_x))$. This gives $\int dz_x dx\, p(x, z_x) \ln(p(x|z_x)) \geq \int dz_x dx\, p(x, z_x) \ln(q(x|z_x))$. We add the entropy of $X$ to both sides to arrive at the variational bound:

$$I^D(X; Z_X) = \int dz_x dx\, p(x, z_x) \ln \frac{p(x|z_x)}{p(x)} \geq \int dz_x dx p(x, z_x) \ln \frac{q(x|z_x)}{p(x)} \equiv \tilde{I}^D(X; Z_X). \tag{6}$$

We further simplify $\tilde{I}^D$ by replacing $p(x)$ by samples, $p(x) \approx \frac{1}{N} \sum_i^N \delta(x - x_i)$ and using the $p(z_x|x)$ that we learned previously from the encoder:

$$\tilde{I}^D(X; Z_X) \approx H(X) + \frac{1}{N} \sum_{i=1}^N \int dz_x p(z_x|x_i) \ln(q(x_i|z_x)). \tag{7}$$

Here $H(X)$ does not depend on $p(z_x|x)$ and, therefore, can be dropped from the loss. The variational version of $I^D(Y; Z_Y)$ is obtained analogously:

$$\tilde{I}^D(Y; Z_Y) \approx H(Y) + \frac{1}{N} \sum_{i=1}^N \int dz_x p(z_y|y_i) \ln(q(y_i|z_y)). \tag{8}$$

## 2.4 VARIATIONAL BOUNDS ON DECODER TERMS NOT ON A LEAF - MINE

The variational bound above cannot be applied to the information terms which do not contain leaves in $G_{\mathrm{decoder}}$. For SIB, this corresponds to the $I^D(Z_X, Z_Y)$ term. This information is maximized. To find

a variational bound such that $I^D(Z_X, Z_Y) \geq \tilde{I}^D(Z_X, Z_Y)$, we use the MINE mutual information estimator (Belghazi et al., 2018), which samples both $Z_X$ and $Z_Y$ from their respective variational encoders. Other mutual information estimators, such as $I_{\text{NCE}}$ (Poole et al., 2019), can be used as long as they are differentiable. For our current application, $I_{\text{MINE}}$ was sufficient. We variationally approximate $p(z_x, z_y)$ as $p(z_x)p(z_y)e^{T(z_x,z_y)}/\mathcal{Z}_{\text{norm}}$, where $\mathcal{Z}_{\text{norm}} = \int dz_x dz_y p(z_x)p(z_y)e^{T(z_x,z_y)}$ is the normalization factor. Here $T(z_x, z_y)$ is parameterized by a neural network that takes in samples of the latent spaces $z_x$ and $z_y$ and returns a single number. We again use the positivity of the Kullback-Leibler divergence, $D_{\text{KL}}(p(z_x, z_y)\|p(z_x)p(z_y)e^{T(z_x,z_y)}/\mathcal{Z}_{\text{norm}}) \geq 0$, which implies $\int dz_x dz_y p(z_x, z_y) \ln(p(z_x, z_y)) \geq \int dz_x dz_y p(z_x, z_y) \ln \frac{p(z_x)p(z_y)e^{T(z_x,z_y)}}{\mathcal{Z}_{\text{norm}}}$. Subtracting $\int dz_x dz_y p(z_x, z_y) \ln(p(z_x)p(z_y))$ from both sides, we find:

$$I^D(Z_X; Z_Y) \geq \int dz_x dz_y p(z_x, z_y) \ln \frac{e^{T(z_x,z_y)}}{\mathcal{Z}_{\text{norm}}} \equiv \tilde{I}^D_{\text{MINE}}(Z_X; Z_Y). \tag{9}$$

### 2.5 Parameterizing the distributions and the reparameterization trick

$H(X)$, $H(Y)$, and $I(X, Y)$ do not depend on $p(z_x|x)$ and $p(z_y|y)$ and are dropped from the loss. Further, we can use any ansatz for the variational distributions we introduced. We choose parametric probability distribution families and learn the nearest distribution in these families consistent with the data. We assume $p(z_x|x)$ is a normal distribution with mean $\mu_{Z_X}(x)$ and a diagonal variance $\Sigma_{Z_X}(x)$. We learn the mean and the log variance as neural networks. We also assume that $q(x|z_x)$ is normal with a mean $\mu_X(z_x)$ and a unit variance. In principle, we could also learn the variance for this distribution, but the approach works well as is. Finally, we assume that $r(z_x)$ is a standard normal distribution. We use the reparameterization trick to produce samples of $z_{xi,j} = z_{x_j}(x_i) = \mu(x_i) + \sqrt{\Sigma_{Z_X}(x_i)}\eta_j$ from $p(z_x|x_i)$, where $\eta_j$ is drawn from a standard normal distribution (Kingma & Welling, 2014). We choose the same types of distributions for the corresponding $z_y$ terms.

To sample from $p(z_x, z_y)$ we use $p(z_x, z_y) = \int dx dy\, p(z_x, z_y, x, y) = \int dx dy\, p(z_x|x)p(z_y|y) \times p(x, y) \approx \frac{1}{N}\sum_{i=1}^N p(z_x|x_i)p(z_y|y_i) = \frac{1}{NM^2}\sum_{i=1}^N (\sum_{j=1}^M \delta(z_x - z_{xi,j}))(\sum_{j=1}^M \delta(z_y - z_{yi,j}))$, where $z_{xi,j} \in p(z_x|x_i)$ and $z_{yi,j} \in p(z_y|x_i)$, and $M$ is the number of new samples being generated. To sample from $p(z_x)p(z_y)$, we generate samples from $p(z_x, z_y)$ and scramble the generated entries $z_x$ and $z_y$, destroying all correlations. With this, the components of the loss function become

$$\tilde{I}^E(X; Z_X) \approx \frac{1}{2N}\sum_{i=1}^N \left[ \text{Tr}(\Sigma_{Z_X}(x_i)) + \|\vec{\mu}_{Z_X}(x_i)\|^2 - k_{Z_X} - \ln\det(\Sigma_{Z_X}(x_i)) \right], \tag{10}$$

$$\tilde{I}^D(X; Z_X) \approx \frac{1}{MN}\sum_{i,j=1}^{N,M} -\frac{1}{2}\|(x_i - \mu_X(z_{xi,j}))\|^2, \tag{11}$$

$$\tilde{I}^D_{\text{MINE}}(Z_X; Z_Y) \approx \frac{1}{M^2N}\sum_{i,j_x,j_y=1}^{N,M,M} \left[ T(z_{xi,j_x}, z_{yi,j_y}) - \ln\mathcal{Z}_{\text{norm}} \right], \tag{12}$$

where $\mathcal{Z}_{\text{norm}} = \mathbb{E}_{z_x \sim p(z_x), z_y \sim p(z_y)}[e^{T(z_x,z_y)}]$, $k_{Z_X}$ is the dimension of $Z_X$, and the corresponding terms for $Y$ are similar. Combining these terms results in the variational loss for DVSIB:

$$L_{\text{DVSIB}} = \tilde{I}^E(X; Z_X) + \tilde{I}^E(Y; Z_Y) - \beta\left( \tilde{I}^D_{\text{MINE}}(Z_X; Z_Y) + \tilde{I}^D(X; Z_X) + \tilde{I}^D(Y; Z_Y) \right). \tag{13}$$

## 3 Deriving other DR methods

The variational bounds used in DVSIB can be used to implement loss functions that correspond to other encoder-decoder graph pairs and hence to other DR algorithms. The simplest is the beta variational auto-encoder. Here $G_{\text{encoder}}$ consists of one term: $X$ compressed into $Z_X$. Similarly $G_{\text{decoder}}$ consists of one term: $X$ decoded from $Z_X$ (see Table 1). Using this simple set of Bayesian networks, we find the variational loss

$$L_{\text{beta-VAE}} = \tilde{I}^E(X; Z_X) - \beta\tilde{I}^D(X; Z_X). \tag{14}$$

Table 1: Method descriptions, variational losses, and the Bayesian Network graphs for each DR method derived in our framework. See Appendix A for details.

| Method Description | $G_{\text{encoder}}$ | $G_{\text{decoder}}$ |
|---|---|---|
| **beta-VAE** (Kingma & Welling, 2014; Higgins et al., 2016): Two independent Variational Autoencoder (VAE) models trained, one for each view, $X$ and $Y$ (only $X$ graphs/loss shown).
$L_{\text{VAE}} = \tilde{I}^E(X; Z_X) - \beta\tilde{I}^D(X; Z_X)$ | $X \to Z_X$ | $X \leftarrow Z_X$ |
| **DVIB** (Alemi et al., 2017): Two bottleneck models trained, one for each view, $X$ and $Y$, using the other view as the supervising signal. (Only $X$ graphs/loss shown).
$L_{\text{DVIB}} = \tilde{I}^E(X; Z_X) - \beta\tilde{I}^D(Y; Z_X)$ | $X - Y$, $X \to Z_X$ | $X - Y$, $Z_X \to Y$ |
| **beta-DVCCA**: Similar to DVIB (Alemi et al., 2017), but with reconstruction of both views. Two models trained, compressing either $X$ or $Y$, while reconstructing both $X$ and $Y$. (Only $X$ graphs/loss shown).
$L_{\text{DVCCA}} = \tilde{I}^E(X; Z_X) - \beta(\tilde{I}^D(Y; Z_X) + \tilde{I}^D(X; Z_X))$
**DVCCA** (Wang et al., 2016): $\beta$-DVCCA with $\beta = 1$. | $X - Y$, $X \to Z_X$ | $X - Y$, $Z_X \to X$, $Z_X \to Y$ |
| **beta-joint-DVCCA**: A single model trained using a concatenated variable $[X, Y]$, learning one latent representation $Z$.
$L_{\text{jDVCCA}} = \tilde{I}^E((X, Y); Z) - \beta(\tilde{I}^D(Y; Z) + \tilde{I}^D(X; Z))$
**joint-DVCCA** (Wang et al., 2016): $\beta$-jDVCCA with $\beta = 1$. | $X - Y$, $X \to Z$, $Y \to Z$ | $X - Y$, $Z \to X$, $Z \to Y$ |
| **beta-DVCCA-private**: Two models trained, compressing either $X$ or $Y$, while reconstructing both $X$ and $Y$, and simultaneously learning private information $W_X$ and $W_Y$. (Only $X$ graphs/loss shown).
$L_{\text{DVCCA-p}} = \tilde{I}^E(X; Z) + \tilde{I}^E(X; W_X) + \tilde{I}^E(Y; W_Y) - \beta(\tilde{I}^D(X; (W_X, Z)) + \tilde{I}^D(Y; (W_Y, Z)))$

**DVCCA-private** (Wang et al., 2016): $\beta$-DVCCA-p with $\beta = 1$. | $X - Y$, $W_X$, $Z$, $W_Y$ | $X - Y$, $W_X$, $Z$, $W_Y$ |
| **beta-joint-DVCCA-private**: A single model was trained using a concatenated variable $[X, Y]$, learning one latent representation $Z$, and simultaneously learning private information $W_X$ and $W_Y$.
$L_{\text{jDVCCA-p}} = \tilde{I}^E((X, Y); Z) + \tilde{I}^E(X; W_X) + \tilde{I}^E(Y; W_Y) - \beta(\tilde{I}^D(X; (W_X, Z)) + \tilde{I}^D(Y; (W_Y, Z)))$

**joint-DVCCA-private** (Wang et al., 2016): $\beta$-jDVCCA-p $\beta = 1$. | $X - Y$, $W_X$, $Z$, $W_Y$ | $X - Y$, $W_X$, $Z$, $W_Y$ |
| **DVSIB**: A symmetric model trained, producing $Z_X$ and $Z_Y$.
$L_{\text{DVSIB}} = \tilde{I}^E(X; Z_X) + \tilde{I}^E(Y; Z_Y)$
$- \beta\left(\tilde{I}^D_{\text{MINE}}(Z_X; Z_Y) + \tilde{I}^D(X; Z_X) + \tilde{I}^D(Y; Z_Y)\right)$ | $X - Y$, $X \to Z_X$, $Y \to Z_Y$ | $X - Y$, $Z_X \to X$, $Z_Y \to Y$, $Z_X \to Z_Y$ |
| **DVSIB-private**: A symmetric model trained, producing $Z_X$ and $Z_Y$, while simultaneously learning private information $W_X$ and $W_Y$.
$L_{\text{DVSIBp}} = \tilde{I}^E(X; W_X) + \tilde{I}^E(X; Z_X) +$
$\tilde{I}^E(Y; Z_Y) + \tilde{I}^E(Y; W_Y) -$
$\beta\left(\tilde{I}^D_{\text{MINE}}(Z_X; Z_Y) + \tilde{I}^D(X; (Z_X, W_X)) + \tilde{I}^D(Y; (Z_Y, W_Y))\right)$ | $X - Y$, $w_X$, $Z_X$, $Z_Y$, $w_Y$ | $X - Y$, $w_X$, $Z_X \to Z_Y$, $w_Y$ |

Both terms in Eq. (14) can be approximated and implemented by neural networks. Similarly, we can re-derive the DVCCA family of losses (Wang et al., 2016). Here $G_{\text{encoder}}$ is $X$ compressed into $Z_X$. $G_{\text{decoder}}$ reconstructs both $X$ and $Y$ from the same compressed latent space $Z_X$. In fact, our loss function is more general than the DVCCA loss and has an additional compression-reconstruction trade-off parameter $\beta$. We call this more general loss $\beta$-DVCCA, and the original DVCCA emerges when $\beta = 1$. Table 1 shows how our framework reproduces and generalizes other DR losses (see Appendix A). Our framework naturally extends beyond two variables as well (see Appendix B).

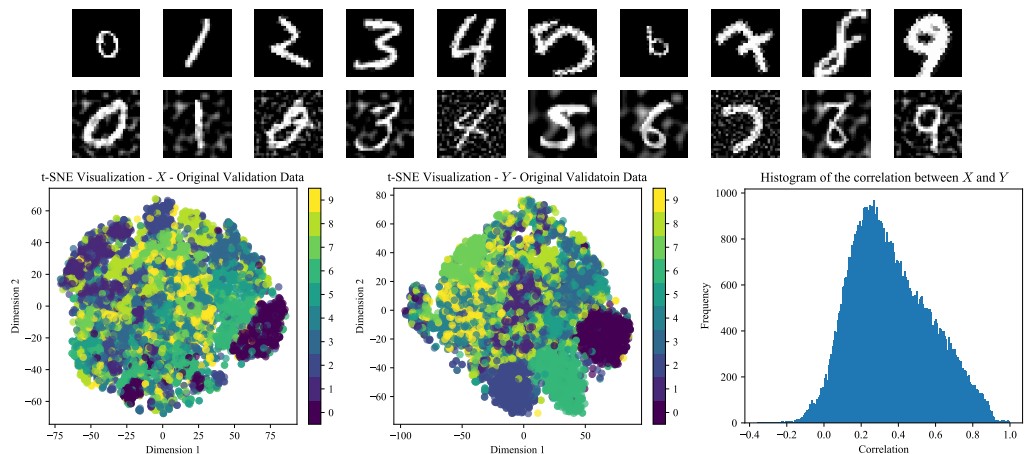

Figure 2: Dataset consisting of pairs of digits drawn from MNIST that share an identity. Top row, $X$: MNIST digits randomly scaled $(0.5 - 1.5)$ and rotated $(0 - \pi/2)$. Bottom row, $Y$: MNIST digits with a background Perlin noise. t-SNE of $X$ and $Y$ datasets (left and middle) shows poor separation by digit, and there is a wide range of correlation between $X$ and $Y$ (right).

## 4 RESULTS

To test our methods, we created a dataset inspired by the noisy MNIST dataset (LeCun et al., 1998; Wang et al., 2015; 2016), consisting of two distinct views of data, both with dimensions of $28 \times 28$ pixels, cf. Fig. 2. The first view comprises the original image randomly rotated by an angle uniformly sampled between 0 and $\frac{\pi}{2}$ and scaled by a factor uniformly distributed between $0.5$ and $1.5$. The second view consists of the original image with an added background Perlin noise (Perlin, 1985) with the noise factor uniformly distributed between 0 and 1. Both image intensities are scaled to the range of $[0, 1)$. To ensure a challenging task, the dataset was shuffled within labels, retaining only the shared label identity between two images, while disregarding the view-specific details, i.e., the random rotation and scaling for $X$, and the background noise for $Y$. The dataset, totaling $70,000$ images, was partitioned into training $(80\%)$, testing $(10\%)$, and validation $(10\%)$ subsets. Visualization via t-SNE (Hinton & Roweis, 2002) plots of the original dataset suggest poor separation by digit, and the two digit views have diverse correlations, making this a sufficiently hard problem.

The DR methods we evaluated include all methods from Tbl. 1. CCA (Hotelling, 1936) served as a baseline for linear dimensionality reduction. Multi-view Information Bottleneck Federici et al. (2020b) was included for a specific comparison with DVSIB (see Appendix C). We emphasize that none of the algorithms were given labeled data. They had to infer compressed latent representations that presumably should cluster into ten different digits based simply on the fact that images come in pairs, and the (unknown) digit label is the only information that relates the two images.

Each method was trained for 100 epochs using fully connected neural networks with layer sizes $(\text{input\_dim}, 1024, 1024, k_Z, k_Z)$, where $k_Z$ is the latent dimension size, employing ReLU activations for the hidden layers. The input dimension (input\_dim) was either the size of $X$ (784) or the size of the concatenated $[X, Y]$ (1568). The last two layers of size $k_Z$ represented the means and $\log(\text{variance})$ learned. For the decoders, we employed regular decoders, fully connected neural networks with layer sizes $(k_Z, 1024, 1024, \text{output\_dim})$, using ReLU activations for the hidden layers and sigmoid activation for the output layer. Again, the output dimension (output\_dim) could either be the size of $X$ (784) or the size of the concatenated $[X, Y]$ (1568). The latent dimension $(k_Z)$ could be $k_{Z_X}$ or $k_{Z_Y}$ for regular decoders, or $k_{Z_X} + k_{W_X}$ or $k_{Z_Y} + k_{W_Y}$ for decoders with private information. Additionally, another decoder denoted as decoder\_MINE, based on the MINE estimator for estimating $I(Z_X, Z_Y)$, was used in DVSIB and DVSIB with private information. The decoder\_MINE is a fully connected neural network with layer sizes $(k_{Z_X} + k_{Z_Y}, 1024, 1024, 1)$ and ReLU activations for the hidden layers. Optimization was conducted using the ADAM optimizer with default parameters.

To evaluate the methods, we trained them on the training portions of $X$ and $Y$ without exposure to the true labels. Subsequently, we utilized the trained encoders to compute $Z_{\text{train}}$, $Z_{\text{test}}$, and $Z_{\text{validation}}$ on the respective datasets. To assess the quality of the learned representations, we revealed the labels of $Z_{\text{train}}$ and trained a linear SVM classifier with $Z_{\text{train}}$ and labels$_{\text{train}}$. Fine-tuning of the classifier was performed to identify the optimal SVM slack parameter ($C$ value), maximizing accuracy on $Z_{\text{test}}$. This best classifier was then used to predict $Z_{\text{validation}}$, yielding the reported accuracy. We also

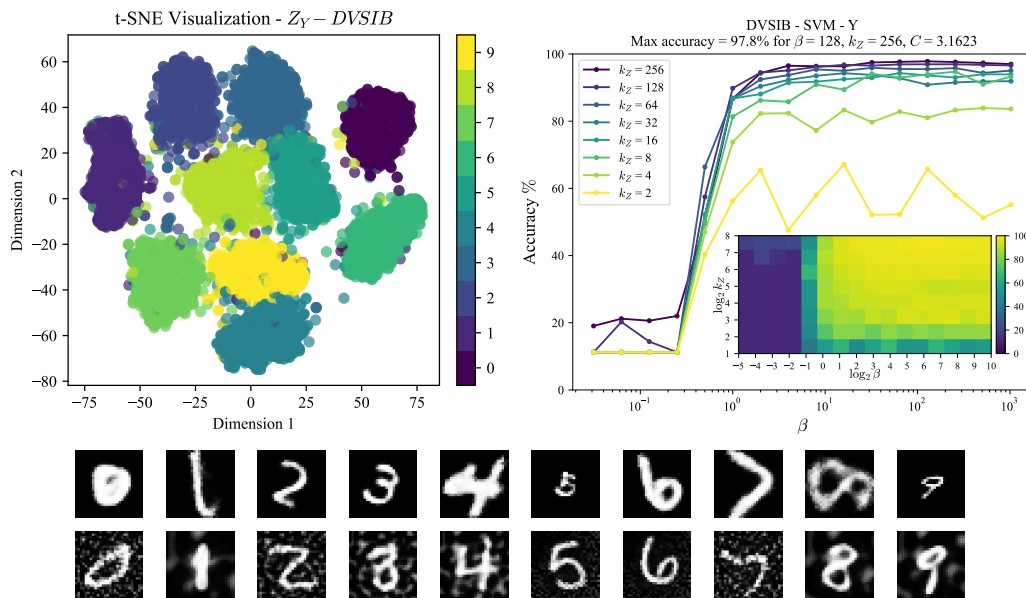

Figure 3: Top: t-SNE plot of the latent space $Z_Y$ of DVSIB colored by the identity of digits. Top Right: Classification accuracy of an SVM trained on DVSIB's $Z_Y$ latent space. The accuracy was evaluated for DVSIB with a parameter sweep of the trade-off parameter $\beta = 2^{-5}, ..., 2^{10}$ and the latent dimension $k_Z = 2^1, ..., 2^8$. The max accuracy was $97.8\%$ for $\beta = 128$ and $k_Z = 256$. Bottom: Example digits generated by sampling from the DVSIB decoder, $X$ and $Y$ branches.

conducted classification experiments using fully connected neural networks, with detailed results available in the Appendix E. For both SVM and the fully connected network, we find the baseline accuracy on the original training data and labels $(X_{\text{train}}, \text{labels}_{\text{train}})$ and $(Y_{\text{train}}, \text{labels}_{\text{train}})$, fine-tuning with the test datasets, and reporting the results of the validation datasets. Here, we focus on the results of the $Y$ datasets (MNIST with correlated noise background); results for $X$ are in the Appendix E. A parameter sweep was performed to identify optimal $k_Z$ values, ranging from $2^1$ to $2^8$ dimensions on $\log_2$ scale, as well as optimal $\beta$ values, ranging from $2^{-5}$ to $2^{10}$. For methods with private information, $k_{W_X}$ and $k_{W_Y}$ were varied from $2^1$ to $2^6$. The highest accuracy is reported in Tbl. 2, along with the optimal parameters used to obtain this accuracy. Additionally, for every method we find the range of $\beta$ and the dimensionality $k_Z$ of the latent variable $Z_Y$ that gives $95\%$ of the method's maximum accuracy. If the range includes the limits of the parameter, this is indicated by an asterisk.

Figure 3 shows a t-SNE plot of DVSIB's latent space, $Z_Y$, colored by the identity of digits. The resulting latent space has 10 clusters, each corresponding to one digit. The clusters are well separated and interpretable. Further, DVSIB's $Z_Y$ latent space provides the best classification of digits using a linear method such as an SVM showing the latent space is linearly separable. DVSIB maximum classification accuracy obtained for the linear SVM is $97.8\%$. Crucially, DVSIB maintains accuracy of at least $92.9\%$ ($95\%$ of $97.8\%$) for $\beta \in [2, 1024^*]$ and $k_Z \in [8, 256^*]$. This accuracy is high compared to other methods and has a large range of hyperparameters that maintain its ability to correctly capture information about the identity of the shared digit. DVSIB is a generative method, we have provided sample generated digits from the decoders that were trained from the model graph.

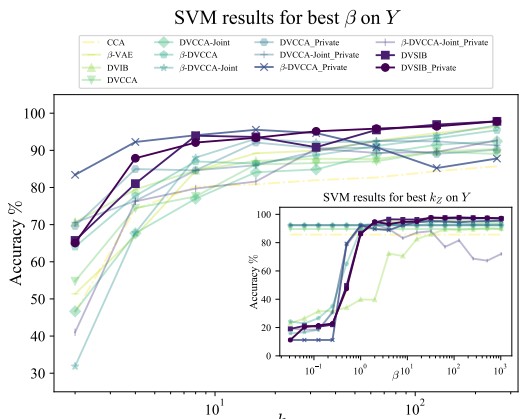

Figure 4: The best SVM classification accuracy curves for each method. Here DVSIB and DVSIB-private obtained the best accuracy and, together with $\beta$-DVCCA-private, they had the best accuracy for low latent dimensional spaces.

In Fig. 4, we show the highest SVM classification accuracy curves for each method. DVSIB and DVSIB-private tie for the best classification accuracy for $Y$. Together with $\beta$-DVCCA-private they have the highest accuracy for all dimensions

of the latent space, $k_Z$. In theory, only one dimension should be needed to capture the identity of a digit, but our data sets also contain information about the rotation and scale for $X$ and the strength of the background noise for $Y$. $Y$ should then need at least two latent dimensions to be reconstructed and $X$ should need at least three. Since DVSIB, DVSIB-private, and $\beta$-DVCCA-private performed with the best accuracy starting with the smallest $k_Z$, we conclude that methods with the encoder-decoder graphs that more closely match the structure of the data produce higher accuracy with lower dimensional latent spaces.

We also evaluate DVSIB on the MIR-FLICKR dataset (Huiskes & Lew, 2008), see Appendix D.

Table 2: Maximum accuracy from a linear SVM and the optimal $k_Z$ and $\beta$ for variational DR methods reported on the $Y$ (above the line) and the joint $[X, Y]$ (below the line) datasets. ($^{\dagger}$ fixed values)

| Method | Acc. % | $k_{Z\,\textbf{best}}$ | 95% $k_{Z\,\textbf{range}}$ | $\beta_{\textbf{best}}$ | 95% $\beta_{\textbf{range}}$ | $C_{\text{best}}$ |
|---|---|---|---|---|---|---|
| Baseline | 90.8 | $784^{\dagger}$ | - | - | - | 0.1 |
| CCA | 85.7 | 256 | [32,256*] | - | - | 10 |
| $\beta$-VAE | 96.3 | 256 | [64,256*] | 32 | [2,1024*] | 10 |
| DVIB | 90.4 | 256 | [16,256*] | 512 | [8,1024*] | 0.003 |
| DVCCA | 89.6 | 128 | [16,256*] | $1^{\dagger}$ | - | 31.623 |
| $\beta$-DVCCA | 95.4 | 256 | [64,256*] | 16 | [2,1024*] | 10 |
| DVCCA-p | 92.1 | 16 | [16,256*] | $1^{\dagger}$ | - | 0.316 |
| $\beta$-DVCCA-p | 95.5 | 16 | [**4**,256*] | 1024 | [1,1024*] | 0.316 |
| MVIB | **97.7** | 8 | [**4**,64] | 1024 | [128,1024] | 0.01 |
| DVSIB | **97.8** | 256 | [**8**,256*] | 128 | [2,1024*] | 3.162 |
| DVSIB-p | **97.8** | 256 | [**8**,256*] | 32 | [2,1024*] | 10 |
| jBaseline | 91.9 | $1568^{\dagger}$ | - | - | - | 0.003 |
| jDVCCA | 92.5 | 256 | [64,265*] | $1^{\dagger}$ | - | 10 |
| $\beta$-jDVCCA | 96.7 | 256 | [16,265*] | 256 | [1,1024*] | 1 |
| jDVCCA-p | 92.5 | 64 | [32,265*] | $1^{\dagger}$ | - | 10 |
| $\beta$-jDVCCA-p | 92.7 | 256 | [**4**,265*] | 2 | [1,1024*] | 10 |

## 5 CONCLUSION

We developed an MIB-based framework for deriving variational loss functions for DR applications. We demonstrated the use of this framework by developing a novel variational method, DVSIB. DVSIB compresses the variables $X$ and $Y$ into latent variables $Z_X$ and $Z_Y$ respectively, while maximizing the information between $Z_X$ and $Z_Y$. This example is particularly illuminating as it shows how to find variational bounds on all types of terms found in all considered DR methods. We provide a library of typical terms in Appendices. We also (re)-derive several DR methods in Tbl. 1. These include well-known techniques such as $\beta$-VAE, DVIB, DVCCA, and DVCCA-private. MIB naturally introduces a trade-off parameter into the DVCCA family of methods, resulting in what we term the $\beta$-DVCCA DR methods, of which DVCCA is a special case. We implement this new family of methods and show that it produces better latent spaces than DVCCA at $\beta = 1$, cf. Tbl. 2.

We observe that methods that more closely match the structure of dependencies in the data can give better latent spaces as measured by the dimensionality of the latent space and the accuracy of reconstruction (see Figure 4). This makes DVSIB, DVSIB-private, and $\beta$-DVCCA-private perform the best. DVSIB and DVSIB-private both have separate latent spaces for $X$ and $Y$. The private methods allow us to learn additional aspects about $X$ and $Y$ that are not important for the shared digit label, but allow reconstruction of the rotation and scale for $X$ and the background noise of $Y$.

Our framework may be extended beyond variational approaches. For instance, in the deterministic limit of VAE, autoencoders can be retrieved by defining the encoder/decoder graphs as nonlinear neural networks $z = f(x)$ and $x = g(z)$. Additionally, linear methods like CCA can be viewed as special cases of the information bottleneck (Chechik et al., 2003) and hence must follow from our approach. Similarly, by using specialized encoder and decoder neural networks, e.g., convolutional ones, our framework can implement symmetries and other constraints into the DR process. Overall, the framework serves as a versatile and customizable toolkit, capable of encompassing a wide spectrum of dimensionality reduction methods. With the provided tools and code, we aim to facilitate adaptation of the approach to diverse problems.

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

# A DERIVING AND DESIGNING VARIATIONAL LOSSES

In the next two sections, we provide a library of typical terms found in encoder graphs, Appendix A.1, and decoder graphs, AppendixA.2. In Appendix A.3, we provide examples of combining these terms to produce variational losses corresponding to beta-VAE, DVIB, beta-DVCCA, beta-DVCCA-joint, beta-DVCCA-private, DVSIB, and DVSIB-private.

## A.1 ENCODER GRAPH COMPONENTS

We expand Sec. 2.2 and present a range of common components found in encoder graphs across various DR methods, cf. Fig. (5).

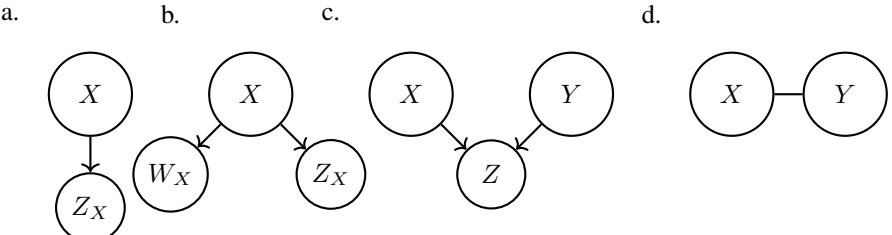

Figure 5: Encoder graph components.

a. This graph corresponds to compressing the random variable $X$ to $Z_X$. Variational bounds for encoders of this type were derived in the main text in Sec. 2.2 and correspond to the loss:

$$\tilde{I}^E(X; Z_X) = \frac{1}{N} \sum_{i=1}^{N} D_{\mathrm{KL}}(p(z_x|x_i)\|r(z_x))$$

$$\approx \frac{1}{2N} \sum_{i=1}^{N} \left[ \mathrm{Tr}(\Sigma_{Z_X}(x_i)) + \|\vec{\mu}_{Z_X}(x_i)\|^2 - k_{Z_X} - \ln \det(\Sigma_{Z_X}(x_i)) \right]. \quad (15)$$

b. This type of encoder graph is similar to the first, but now with two outputs, $Z_X$ and $W_X$. This corresponds to making two encoders, one for $Z_X$ and one for $W_X$, $\tilde{I}^E(Z_X; X) + \tilde{I}^E(W_X; X)$, where

$$\tilde{I}^E(Z_X; X) \approx \frac{1}{N} \sum_{i=1}^{N} D_{\mathrm{KL}}(p(z_x|x_i)\|r(z_x)), \quad (16)$$

$$\tilde{I}^E(W_X; X) \approx \frac{1}{N} \sum_{i=1}^{N} D_{\mathrm{KL}}(p(w_x|x_i)\|r(w_x)). \quad (17)$$

c. This type of encoder consists of compressing $X$ and $Y$ into a single variable $Z$. It corresponds to the information loss $I^E(Z; (X, Y))$. This again has a similar encoder structure to type (a), but $X$ is replaced by a joint variable $(X, Y)$. For this loss, we find a variational version:

$$\tilde{I}^E(Z; (X, Y)) \approx \frac{1}{N} \sum_{i=1}^{N} D_{\mathrm{KL}}(p(z|x_i, y_i)\|r(x_i, y_i)). \quad (18)$$

d. This final type of an encoder term corresponds to information $I^E(X, Y)$, which is constant with respect to our minimization. In practice, we drop terms of this type.

## A.2 DECODER GRAPH COMPONENTS

In this section, we elaborate on the decoder graphs that happen in our considered DR methods, cf. Fig. (6).

All decoder graphs sample from their methods' corresponding encoder graph.

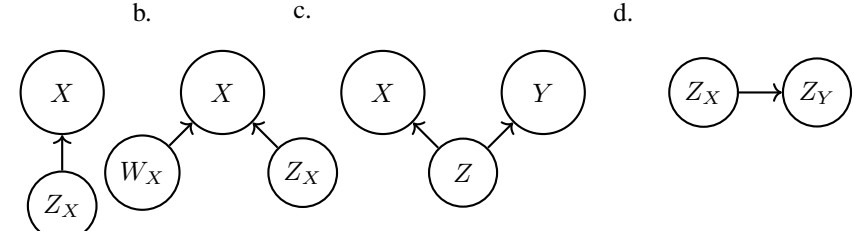

Figure 6: Decoder graph components.

a. In this decoder graph, we decode $X$ from the compressed variable $Z_X$. Variational bounds for decoders of this type were derived in the main text, Sec. 2.3, and they correspond to the loss:

$$\tilde{I}^D(X; Z_X) = H(X) + \frac{1}{N} \sum_{i=1}^{N} \int dz_x p(z_x|x_i) \ln q(x_i|z_x)$$

$$\approx H(X) + \frac{1}{MN} \sum_{i,j=1}^{N,M} -\frac{1}{2} ||(x_i - \mu_X(z_{xi,j}))||^2, \quad (19)$$

where $H(X)$ can be dropped from the loss since it doesn't change in optimization.

b. This type of decoder term is similar to that in part (a), but $X$ is decoded from two variables simultaneously. The corresponding loss term is $I^D(X; (Z_X, W_X))$. We find a variational loss by replacing $Z_X$ in part (a) by $(Z_X, W_X)$:

$$\tilde{I}^D(X; (Z_X, W_X)) \approx H(X) + \frac{1}{N} \sum_{i=1}^{N} \int dz_x dw_x p(z_x, w_x|x_i) \ln(q(x_i|z_x, w_x)), \quad (20)$$

where, again, the entropy of $X$ can be dropped.

c. This decoder term can be obtained by adding two decoders of type (a) together. In this case, the loss term is $I^D(X; Z) + I^D(Y; Z)$:

$$\tilde{I}^D(X; Z) + \tilde{I}^D(Y; Z) \approx H(X) + H(Y)$$

$$+ \frac{1}{N} \sum_{i=1}^{N} \int dz p(z|x_i) \ln(q(x_i|z)) + \frac{1}{N} \sum_{i=1}^{N} \int dz p(z|y_i) \ln(q(y_i|z)), \quad (21)$$

and the entropy terms can be dropped, again.

d. Decoders of this type were discussed in the main text in Sec. 2.4. They correspond to the information between latent variables $Z_X$ and $Z_Y$. We use the MINE estimator to find variational bounds for such terms:

$$\tilde{I}^D_{\text{MINE}}(Z_X; Z_Y) = \int dz_x dz_y p(z_x, z_y) \ln \frac{e^{T(z_x,z_y)}}{\mathcal{Z}_{\text{norm}}} \approx \frac{1}{NM^2} \sum_{i,j_x,j_y=1}^{N,M,M} \left[ T(z_{xi,j_x}, z_{yi,j_y}) - \ln \mathcal{Z}_{\text{norm}} \right].$$

$$(22)$$

## A.3 DETAILED METHOD IMPLEMENTATIONS

For completeness, we provide detailed implementations of methods outlined in Tbl. 1.

### A.3.1 BETA VARIATIONAL AUTO-ENCODER

A variational autoencoder (Kingma & Welling, 2014; Higgins et al., 2016) compresses $X$ into a latent variable $Z_X$ and then reconstructs $X$ from the latent variable, cf. Fig. (8). The overall loss is a

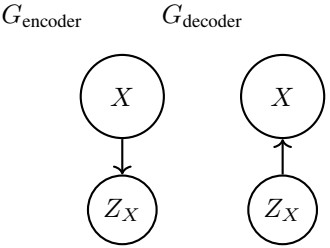

Figure 7: Encoder and decoder graphs for the beta-variational auto-encoder method

trade-off between the compression $I^E(X; Z_X)$ and the reconstruction $I^D(X; Z_X)$:

$$I^E(X; Z_X) - \beta I^D(X; Z_X) \leq \tilde{I}^E(X; Z_X) - \beta \tilde{I}^D(X; Z_X)$$

$$\lesssim \frac{1}{N} \sum_{i=1}^{N} D_{\text{KL}}(p(z_x|x_i)\|r(z_x)) - \beta \left( H(X) + \frac{1}{N} \sum_{i=1}^{N} \int dz_x p(z_x|x_i) \ln(q(x_i|z_x)) \right). \quad (23)$$

$H(X)$ is a constant with respect to the minimization, and it can be omitted from the loss. Similar to the main text, DVSIB case, we make ansatzes for forms of each of the variational distributions. We choose parametric distribution families and learn the nearest distribution in these families consistent with the data. Specifically, we assume $p(z_x|x)$ is a normal distribution with mean $\mu_{Z_X}(X)$ and variance $\Sigma_{Z_X}(X)$. We learn the mean and the log-variance as neural networks. We also assume that $q(x|z_x)$ is normal with a mean $\mu_X(z_x)$ and a unit variance. Finally, we assume that $r(z_x)$ is drawn from a standard normal distribution. We then use the re-parameterization trick to produce samples of $z_{x_j}(x) = \mu(x) + \sqrt{\Sigma_{Z_X}(x)}\eta_j$ from $p(z_x|x)$, where $\eta$ is drawn from a standard normal distribution. Overall, this gives:

$$L_{\text{VAE}} = \frac{1}{2N} \sum_{i=1}^{N} \left[ \text{Tr}(\Sigma_{Z_X}(x_i)) + \vec{\mu}_{Z_X}(x_i)^T \vec{\mu}_{Z_X}(x_i) - k_{Z_X} - \ln \det(\Sigma_{Z_X}(x_i)) \right]$$

$$- \beta \left( \frac{1}{MN} \sum_{i=1}^{N} \sum_{j=1}^{M} -\frac{1}{2}(x_i - \mu_X(z_{x_j}))^T (x_i - \mu_X(z_{x_j})) \right). \quad (24)$$

This is the same loss as for a beta auto-encoder. However, following the convention in the Information Bottleneck literature (Tishby et al., 2000; Friedman et al., 2013), our $\beta$ is the inverse of the one typically used for beta auto-encoders. A small $\beta$ in our case results in a stronger compression, while a large $\beta$ results in a better reconstruction.

### A.3.2 DEEP VARIATIONAL INFORMATION BOTTLENECK

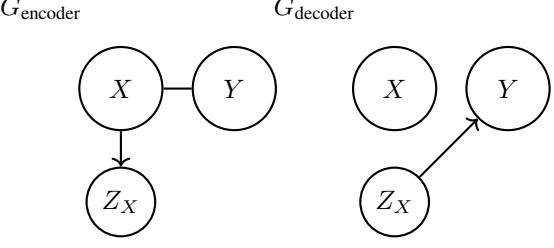

Figure 8: Encoder and decoder graphs for the Deep Variational Information Bottleneck.

Just as in the beta auto-encoder, we immediately write down the loss function for the information bottleneck. Here, the encoder graph compresses $X$ into $Z_X$, while the decoder tries to maximize the

information between the compressed variable and the relevant variable $Y$, cf. Fig. (8). The resulting loss function is:

$$L_{\text{IB}} = I^E(X;Y) + I^E(X;Z_X) - \beta I^D(Y;Z_X). \tag{25}$$

Here the information between $X$ and $Y$ does not depend on $p(z_x|x)$ and can dropped in the optimization.

Thus the Deep Variational Information Bottleneck (Alemi et al., 2017) becomes :

$$L_{\text{DVIB}} \quad \approx \quad \frac{1}{N}\sum_{i=1}^{N} D_{\text{KL}}(p(z_x|x_i)\|r(z_x)) \;-\; \beta\left(\frac{1}{N}\sum_{i=1}^{N}\int dz_x p(z_x|x_i)\ln(q(y_i|z_x))\right), \tag{26}$$

where we dropped $H(Y)$ since it doesn't change in the optimization.

As we have been doing before, we choose to parameterize all these distributions by Gaussians and their means and their log variances are learned by neural networks. Specifically, we parameterize $p(z_x|x) = N(\mu_{z_x}(x), \Sigma_{z_x})$, $r(z_x) = N(0, I)$, and $q(y|z_x) = N(\mu_Y, I)$. Again we can use the reparameterization trick and sample from $p(z_x|x_i)$ by $z_{x_j}(x) = \mu(x) + \sqrt{\Sigma_{z_x}(x)}\eta_j$ where $\eta$ is drawn from a standard normal distribution.

### A.3.3  Beta Deep Variational CCA

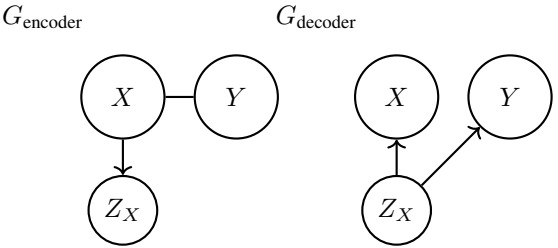

$G_{\text{encoder}}$ $\qquad\qquad$ $G_{\text{decoder}}$

Figure 9: Encoder and decoder graphs for beta Deep Variational CCA.

beta-DVCCA, cf. Fig. 9, is similar to the traditional information bottleneck, but now $X$ and $Y$ are both used as relevance variables:

$$L_{\text{DVCCA}} = \tilde{I}^E(X;Y) + \tilde{I}^E(X;Z_X) - \beta(\tilde{I}^D(Y;Z_X) + \tilde{I}^D(X;Z_X)) \tag{27}$$

Using the same library of terms as before, we find:

$$L_{\text{DVCCA}} \approx \frac{1}{N}\sum_{i=1}^{N} D_{\text{KL}}(p(z_x|x_i)\|r(z_x))$$
$$- \beta\left(\frac{1}{N}\sum_{i=1}^{N}\int dz_x p(z_x|x_i)\ln(q(y_i|z_x)) + \frac{1}{N}\sum_{i=1}^{N}\int dz_x p(z_x|x_i)\ln(q(x_i|z_x))\right). \tag{28}$$

This is similar to the loss function of the deep variational CCA (Wang et al., 2016), but now it has a trade-off parameter $\beta$. It trades off the compression into $Z$ against the reconstruction of $X$ and $Y$ from the compressed variable $Z$.

### A.3.4  beta joint-Deep Variational CCA

Joint deep variational CCA (Wang et al., 2016), cf. Fig. 10, compresses $(X, Y)$ into one $Z$ and then reconstructs the individual terms $X$ and $Y$,

$$L_{\text{DVCCA}} = I^E(X;Y) + I^E((X,Y);Z) - \beta(I^D(Y;Z) + I^D(X;Z)). \tag{29}$$

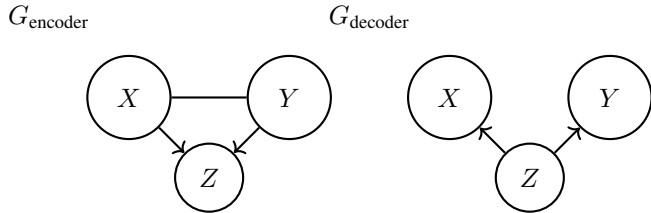

Figure 10: Encoder and decoder graphs for beta joint-Deep Variational CCA.

Using the terms we derived, the loss function is:

$$L_{\text{DVCCA}} \approx \frac{1}{N}\sum_{i=1}^{N} D_{KL}(p(z|x_i, y_i)\|r(z))$$
$$- \beta\left(\frac{1}{N}\sum_{i=1}^{N}\int dz p(z|x_i)\ln(q(y_i|z)) + \frac{1}{N}\sum_{i=1}^{N}\int dz p(z|x_i)\ln(q(x_i|z))\right). \quad (30)$$

The information between $X$ and $Y$ does not change under the minimization and can be dropped.

### A.3.5 BETA (JOINT) DEEP VARIATIONAL CCA-PRIVATE

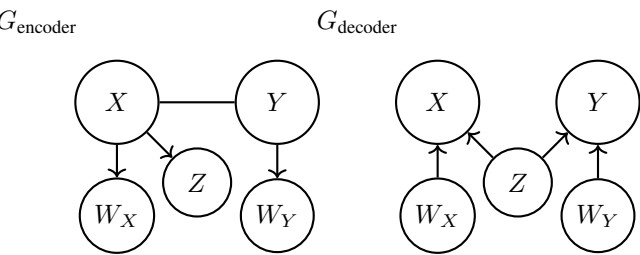

Figure 11: Encoder and decoder graphs for beta Deep Variational CCA-private

This is a generalization of the Deep Variational CCA Wang et al. (2016) to include private information, cf. Fig. 11. Here $X$ is encoded into a shared latent variable $Z$ and a private latent variable $W_X$. Similarly $Y$ is encoded into the same shared variable and a different private latent variable $W_Y$. $X$ is reconstructed from $Z$ and $W_X$, and $Y$ is reconstructed from $Z$ and $W_Y$. In the joint version $(X, Y)$ are compressed jointly in $Z$ similar to the previous joint methods. What follows is the loss $X$ version of beta Deep Variational CCA-private.

$$L_{\text{DVCCAp}} = I^E(X;Y) + I^E((X,Y);Z) + I^E(X;W_X) + I^E(Y;W_Y)$$
$$- \beta(I^D(X;(W_X,Z)) + I^D(Y;(W_Y,Z))). \quad (31)$$

After the usual variational manipulations, this becomes:

$$L_{\text{DVCCAp}} \approx \frac{1}{N}\sum_{i=1}^{N} D_{\text{KL}}(p(z|x_i)\|r(z)) + \frac{1}{N}\sum_{i=1}^{N} D_{\text{KL}}(p(w_x|x_i)\|r(w_x))$$
$$+ \frac{1}{N}\sum_{i=1}^{N} D_{\text{KL}}(p(w_y|y_i)\|r(w_y)) - \beta\left(\frac{1}{N}\sum_{i=1}^{N}\int dz dw_x p(w_x|x_i)p(z|x_i)\ln(q(y_i|z,w_x))\right.$$
$$\left. + \frac{1}{N}\sum_{i=1}^{N}\int dz dw_y p(w_y|y_i)p(z|x_i)\ln(q(x_i|z,w_y))\right). \quad (32)$$

### A.3.6 DEEP VARIATIONAL SYMMETRIC INFORMATION BOTTLENECK

This has been analyzed in detail in the main text, Sec. 2.1, and will not be repeated here.

### A.3.7 DEEP VARIATIONAL SYMMETRIC INFORMATION BOTTLENECK-PRIVATE

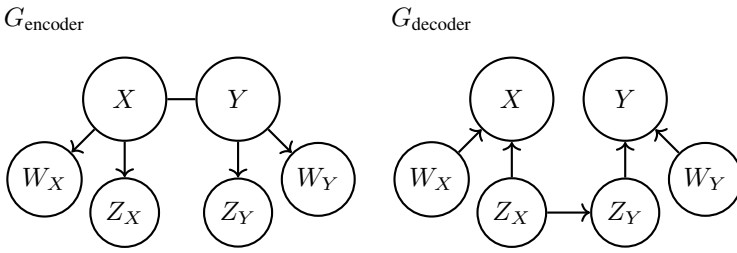

Figure 12: Encoder and decoder graphs for DVSIB-private.

This is a generalization of the Deep Variational Symmetric Information Bottleneck to include private information. Here $X$ is encoded into a shared latent variable $Z_X$ and a private latent variable $W_X$. Similarly $Y$ is encoded into its own shared $Z_Y$ variable and a private latent variable $W_Y$. $X$ is reconstructed from $Z_X$ and $W_X$, and $Y$ is reconstructed from $Z_Y$ and $W_Y$. $Z_X$ and $Z_Y$ are constructed to be maximally informative about each another. This results in

$$L_{\text{DVSIBp}} = I^E(X; W_X) + I^E(X; Z_X) + I^E(Y; Z_Y) + I^E(Y; W_Y)$$
$$- \beta \left( I^D(Z_X; Z_Y) + I^D(X; (Z_X, W_X)) + I^D(Y; (Z_Y, W_Y)) \right). \quad (33)$$

After the usual variational manipulations, this becomes (see also main text):

$$L_{\text{DVSIBp}} \approx \frac{1}{N} \sum_{i=1}^{N} D_{\text{KL}}(p(z_x|x_i)\|r(z_x)) + \frac{1}{N} \sum_{i=1}^{N} D_{\text{KL}}(p(z_y|x_i)\|r(z_y))$$

$$+ \frac{1}{N} \sum_{i=1}^{N} D_{\text{KL}}(p(w_x|x_i)\|r(w_x)) + \frac{1}{N} \sum_{i=1}^{N} D_{\text{KL}}(p(w_y|y_i)\|r(w_y))$$

$$- \beta \left( \int dz_x dz_y p(z_x, z_y) \ln \frac{e^{T(z_x, z_y)}}{\mathcal{Z}_{\text{norm}}} + \frac{1}{N} \sum_{i=1}^{N} \int dz_y dw_y p(w_y|y_i)p(z_y|y_i) \ln(q(y_i|z_y, w_y)) \right.$$

$$\left. + \frac{1}{N} \sum_{i=1}^{N} \int dz_x dw_x p(w_x|x_i)p(z_x|x_i) \ln(q(x_i|z_x, w_x)) \right), \quad (34)$$

where

$$\mathcal{Z}_{\text{norm}} = \int dz_x dz_y p(z_x)p(z_y)e^{T(z_x, z_y)}. \quad (35)$$

## B MULTI-VARIABLE LOSSES (MORE THAN 2 VIEWS / VARIABLES)

It is possible to rederive several multi-variable losses that have appeared in the literature within our framework.

### B.1 MULTI-VIEW TOTAL CORRELATION AUTO-ENCODER

Here we demonstrate several graphs for multi-variable losses. This first example consists of a structure, where all the views $X_1$, $X_2$, and $X_3$ are compressed into the same latent variable $Z$. The corresponding decoder produces reconstructed views from the same latent variable $Z$. This is known in the literature as a multi-view auto-encoder.

$$L_{\text{MVAE}} = \tilde{I}^E((X_1, X_2, X_3); Z) - \beta(\tilde{I}^D(X_1; Z) + \tilde{I}^D(X_2; Z) + \tilde{I}^D(X_3; Z)). \quad (36)$$

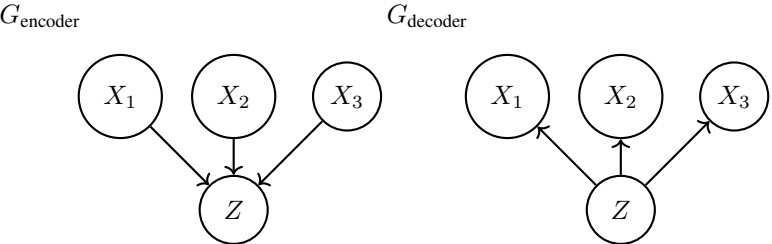

Figure 13: Encoder and decoder graphs for a multi-view auto-encoder.

Using the same library of terms as before, we find:

$$L_{\text{MVAE}} \approx \frac{1}{N} \sum_{i=1}^{N} D_{\text{KL}}(p(z|x_{1i}, x_{2i}, x_{3i}) \| r(z))$$
$$- \beta \left( \frac{1}{N} \sum_{i=1}^{N} \int dz p(z|x_{1i}, x_{2i}, x_{3i}) \ln(q(x_{1i}|z)) + \frac{1}{N} \sum_{i=1}^{N} \int dz p(z|x_{1i}, x_{2i}, x_{3i}) \ln(q(x_{2i}|z)) \right.$$
$$\left. + \frac{1}{N} \sum_{i=1}^{N} \int dz p(z|x_{1i}, x_{2i}, x_{3i}) \ln(q(x_{3i}|z)) \right). \quad (37)$$

## B.2 DEEP VARIATIONAL MULTIMODAL INFORATION BOTTLENECKS

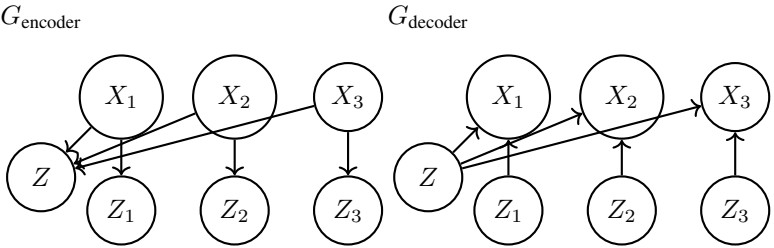

Figure 14: Encoder and decoder graphs for Multimodal Inforation Bottleneck.

This example consists of a structure where all the views $X_1$, $X_2$, and $X_3$ are compressed into separate latent views $Z_1$, $Z_2$, and $Z_3$ and one global shared latent variable $Z$. This structure is analogous to DVCCA-private, but it extends to three variables rather than two. It appears in the literature with slightly different variations. In the decoder graph, $X_1$ is reconstructed from both $Z$ and $Z_1$, $X_2$ is reconstructed from both $Z$ and $Z_2$, and $X_3$ is reconstructed from both $Z$ and $Z_3$.

$$L_{\text{DVAE}} = \tilde{I}^E((X_1, X_2, X_3); Z) + \tilde{I}^E(X_1; Z_1) + \tilde{I}^E(X_2; Z_2) + \tilde{I}^E(X_3; Z_3)$$
$$- \beta(\tilde{I}^D(X_1; (Z, Z_1)) + \tilde{I}^D(X_2; (Z, Z_2)) + \tilde{I}^D(X_3; (Z, Z_3))) \quad (38)$$

Using the same library of terms as before, we find:

$$L_{\text{DVAE}} \approx \frac{1}{N}\sum_{i=1}^{N} D_{\text{KL}}(p(z|x_{1_i}, x_{2_i}, x_{3_i})\|r(z)) + \sum_{j=1}^{3}\frac{1}{N}\sum_{i=1}^{N} D_{\text{KL}}(p(z_j|x_{j_i})\|r_j(z))$$

$$- \beta \left( \frac{1}{N}\sum_{i=1}^{N}\int dz dz_1 dz_2 dz_3 p(z|x_{1_i}, x_{2_i}, x_{3_i})p(z_1|x_{1_i})p(z_2|x_{2_i})p(z_3|x_{3_i})\ln(q(x_{1_i}|z, z_1)) \right.$$

$$+ \frac{1}{N}\sum_{i=1}^{N}\int dz dz_1 dz_2 dz_3 p(z|x_{1_i}, x_{2_i}, x_{3_i})p(z_1|x_{1_i})p(z_2|x_{2_i})p(z_3|x_{3_i})\ln(q(x_{2_i}|z, z_2))$$

$$\left. + \frac{1}{N}\sum_{i=1}^{N}\int dz dz_1 dz_2 dz_3 p(z|x_{1_i}, x_{2_i}, x_{3_i})p(z_1|x_{1_i})p(z_2|x_{2_i})p(z_3|x_{3_i})\ln(q(x_{3_i}|z, z_3)) \right). \quad (39)$$

### B.3 DISCUSSION

There exist many other structures that have been explored in the multi-view representation learning literature, including conditional VIB (Shi et al., 2019; Hwang et al., 2021), which is formulated in terms of conditional information. These types of structures are beyond the current scope of our framework. However, they could be represented by an encoder mapping from all independent views $X_\nu$ to $Z$, subtracted from another encoder mapping from the joint view $\vec{X}$ to $Z$. Coupled with this would be a decoder mapping from $Z$ to the independent views $X_\nu$ (or the joint view $\vec{X}$, analogous to the Joint-DVCCA). Similarly, one can use our framework to represent other multi-view approaches, or their approximations (Lee & Van der Schaar, 2021; Wan et al., 2021; Hwang et al., 2021). This underscores the breadth of methods seeking to address specific questions by exploring known or assumed statistical dependencies within data, and also generality of our approach, which can re-derive these methods.

## C MULTI-VIEW INFORMATION BOTTLENECK

The multiview information bottleneck (MVIB) (Federici et al., 2020b) attempts to remove redundant information between views $(v_1, v_2)$. This is achieved with the following losses:

$$L_1 = I(z_1; v_1|v_2) - \lambda_1 I(v_2; z_1), \quad (40)$$
$$L_2 = I(z_2; v_2|v_1) - \lambda_1 I(v_1; z_2). \quad (41)$$

These losses are equivalent to two deep variational information bottlenecks performed in parallel. Within our framework, the same algorithm emerges with the encoder graph that compresses $v_1$ into $z_1$ and $v_2$ into $z_2$, while the decoder graph would reconstruct $v_2$ from $z_1$ and $v_1$ from $z_2$.

Federici et al. (2020b) combines these two losses while enforcing the condition that $z_1$ and $z_2$ are the same. They bounded the combined loss function to obtain:

$$L_{\text{MVIB}} = D_{SKL}(P(z_1|v_1)\|P(z_2|v_2)) - \beta I(z_1, z_2), \quad (42)$$

with $z_1$ and $z_2$ being the same latent space in this approximation. Here $D_{\text{SKL}}$ is the symmetrized KL divergence, $v_i$ correspond to the two different views, and $z_i$ correspond to their two latent, compressed representation. (Here we changed the parameter $\beta$ to be in front of $I(z_1, z_2)$, to be consistent with the definition of $\beta$ we use elsewhere in this work.) While this loss looks similar to the DVSIB loss, it is conceptually different. It attempts to produce latent variables that are as similar to one another as possible (ideally, $z_1 = z_2$). In contrast, DVSIB attempts to produce different latent variables that could, in theory, have different units, dimensionalities, and domains, while still being as informative about each other as possible. For example, in the noisy MNIST, $Z_X$ contains information about the labels, the angles, and the scale of images (all needed for reconstructing $X$) and no information about the noise structure. At the same time, $Z_Y$ contains information about the labels and the noise factor only (both needed to reconstruct $Y$). See Appendix E.4 for 2-d latent spaces colored by these variables, illustrating the difference between $Z_X$ and $Z_Y$ in DVSIB. Further, in practice, implementation of MVIB uses the same encoder for both views of the data; this is equivalent

to encoding different views using the same function and then trying to force the output to be as close as possible to each other, in contrast to DVSIB.

We evaluate MVIB on the noisy MNIST dataset and include it in Table 2. The performance is similar to that of DVSIB, but slightly worse.

Moreover, MVIB appears to be highly sensitive to parameters and training conditions. Despite employing identical initial conditions and parameters used for training other methods, the approach often experienced collapses during training, resulting in infinities. Interestingly enough, in instances where training persisted for a limited set of parameters (usually low $k_Z$ and high $\beta$), MVIB generated good latent spaces, evidenced by their relatively high classification accuracy.

## D  MIR-FLICKR DATASET RESULT

The MIR-Flickr dataset (Huiskes & Lew, 2008) consists of one million images and 800,000 tags. Each image consist of 3857 hand-crafted features and the 2000 most frequent tags are encoded into a 2000-dimensional binary vector indicating presence or absence of the tag for an image. The dataset is further divided up into 25,000 images which are annotated as belonging to 38 possible topics along with their tags. The remaining 975K image-tag pairs are considered unlabelled. On average, an image belongs to 4.7 topic categories. Following previous literature (Sohn et al., 2014; Wang et al., 2016; Federici et al., 2020b), we remove images that have fewer than two tags resulting in about 750 thousand image-tag pairs.

We evaluate DVSIB on the MIR-Flickr dataset using a standard procedure outlined in the literature Sohn et al. (2014); Wang et al. (2016); Federici et al. (2020b). We train DVSIB on the unlabeled pairs of images and tags to produce two latent spaces, one for images and the other for tags. We then use the trained encoders to encode the labeled set of images, tags, and topics into the latent spaces trained on unlabeled data. The trained encoders output two 64 dimensional latent spaces (one for the image and one for the tags). We then verify how meaningful the representations learned on unlabeled data are for the labeled data. For this, we subdivide the set of 25,000 labeled images into train, test, and validation sets of 10K, 5K, and 10K images, respectively. We train and test a multi-label logistic classifier from the latent representations of the data to the topics. We then evaluate the mean precision of the classifiers based on the latent image space and the latent tag space, both learned from unlabeled data, on the labeled validation data subset. The image latent space has a mean precision of 0.682 and the tag latent space has a mean precision of 0.712. This compares to a reported precision of 0.529 for CCA, 0.565 for Contrastive, 0.573 for DCCA, 0.751 for MV-InfoMax, and 0.749 for MVIB (Wang et al., 2016; Federici et al., 2020b). Note that the precision for the latter two methods is higher than for DVSIB. However, this was achieved with much larger dimensionality of the latent spaces (1024 to 64 in DVSIB), fine-tuning the trade-off parameter $\beta$, deeper networks for variational approximations (4 hidden layers to 1 in our work), as well as fine-tuning many parameters of the optimization algorithms. The performance of the simple, not fine-tuned DVSIB close to these well-tuned methods implemented with much larger networks is remarkable.

## E  ADDITIONAL MNIST RESULTS

In this section, we present supplementary results derived from the methods in Tbl 1.

### E.1  ADDITIONAL RESULTS TABLES FOR THE BEST PARAMETERS

We report classification accuracy using SVM on data $X$, and using neural networks on both $X$ and $Y$.

Table 3: Maximum accuracy from a linear SVM and the optimal $k_Z$ and $\beta$ for variational DR methods on the $X$ dataset. ($^\dagger$ fixed values)

| Method | Acc. % | $k_{Z\,\text{best}}$ | 95% $k_{Z\,\text{range}}$ | $\beta_{\text{best}}$ | 95% $\beta_{\text{range}}$ | $C_{\text{best}}$ |
|---|---|---|---|---|---|---|
| Baseline | 57.8 | $784^\dagger$ | - | - | - | 0.01 |
| CCA | 54.4 | 256 | [8,265*] | - | - | 0.032 |
| $\beta$-VAE | 84.4 | 256 | [128,265*] | 4 | [2,8] | 10 |
| DVIB | 87.3 | 128 | [4,265*] | 512 | [8,1024*] | 0.032 |
| DVCCA | 86.1 | 256 | [64,265*] | $1^\dagger$ | - | 31.623 |
| $\beta$-DVCCA | 88.9 | 256 | [128,265*] | 4 | [1,128] | 10 |
| DVCCA-private | 85.3 | 128 | [32,265*] | $1^\dagger$ | - | 31.623 |
| $\beta$-DVCCA-private | 85.3 | 128 | [32,265*] | 1 | [1,8] | 31.623 |
| MVIB | **93.8** | **8** | [8,16] | 128 | [128,1024*] | 0.01 |
| DVSIB | **92.9** | 256 | [64,265*] | 256 | [4,1024*] | 1 |
| DVSIB-private | **92.6** | 256 | [**32**,265*] | 128 | [8,1024*] | 3.162 |

Table 4: Maximum accuracy from a feed forward neural network and the optimal $k_Z$ and $\beta$ for variational DR methods on the $Y$ and the joined $[X, Y]$ datasets. ($^\dagger$ fixed values)

| Method | Acc. % | $k_{Z\,\text{best}}$ | 95% $k_{Z\,\text{range}}$ | $\beta_{\text{best}}$ | 95% $\beta_{\text{range}}$ |
|---|---|---|---|---|---|
| Baseline | 92.8 | $784^\dagger$ | - | - | - |
| CCA | 90.2 | 256 | [32,256*] | - | - |
| $\beta$-VAE | **98.4** | 64 | [8,256*] | 64 | [2,1024*] |
| DVIB | 90.4 | 128 | [8,256*] | 1024 | [8,1024*] |
| DVCCA | 91.3 | 16 | [4,256*] | $1^\dagger$ | - |
| $\beta$-DVCCA | 97.5 | 128 | [8,256*] | 512 | [2,1024*] |
| DVCCA-private | 93.8 | 16 | [2,256*] | $1^\dagger$ | - |
| $\beta$-DVCCA-private | 97.5 | 256 | [**2**,256*] | 32 | [1,1024*] |
| MVIB | 97.5 | 16 | [8,16] | 256 | [128,1024*] |
| DVSIB | **98.3** | 256 | [**4**,256*] | 32 | [2,1024*] |
| DVSIB-private | **98.3** | 256 | [**4**,256*] | 32 | [2,1024*] |
| Baseline-joint | 97.7 | $1568^\dagger$ | - | - | - |
| joint-DVCCA | 93.7 | 256 | [8,256*] | $1^\dagger$ | - |
| $\beta$-joint-DVCCA | **98.9** | 64 | [8,256*] | 512 | [2,1024*] |
| joint-DVCCA-private | 93.5 | 16 | [4,256*] | $1^\dagger$ | - |
| $\beta$-joint-DVCCA-private | 95.6 | 32 | [4,256*] | 512 | [1,1024*] |

Table 5: Maximum accuracy from a neural network the optimal $k_Z$ and $\beta$ for variational DR methods on the $X$ dataset. ($^\dagger$ fixed values)

| Method | Acc. % | $k_{Z\,\text{best}}$ | 95% $k_{Z\,\text{range}}$ | $\beta_{\text{best}}$ | 95% $\beta_{\text{range}}$ |
|---|---|---|---|---|---|
| Baseline | 92.8 | $784^\dagger$ | - | - | - |
| CCA | 72.6 | 256 | [256,256*] | - | - |
| $\beta$-VAE | 93.3 | 256 | [16,256*] | 256 | [2,1024*] |
| DVIB | 87.5 | 4 | [2,256*] | 1024 | [4,1024*] |
| DVCCA | 87.5 | 128 | [8,256*] | $1^\dagger$ | - |
| $\beta$-DVCCA | 92.2 | 64 | [8,256*] | 32 | [2,1024*] |
| DVCCA-private | 88.2 | 8 | [8,256*] | $1^\dagger$ | - |
| $\beta$-DVCCA-private | 90.7 | 256 | [4,256*] | 8 | [1,1024*] |
| MVIB | **93.6** | 8 | [**8**,16] | 256 | [128,1024*] |
| DVSIB | **93.9** | 128 | [**8**,256*] | 16 | [2,1024*] |
| DVSIB-private | 92.8 | 32 | [8,256*] | 256 | [4,1024*] |

## E.2 t-SNE Embeddings at best parameters

Figures 15 and 16 display display 2d t-SNE embeddings for variables $Z_X$ and $Z_Y$ generated by various considered DR methods.

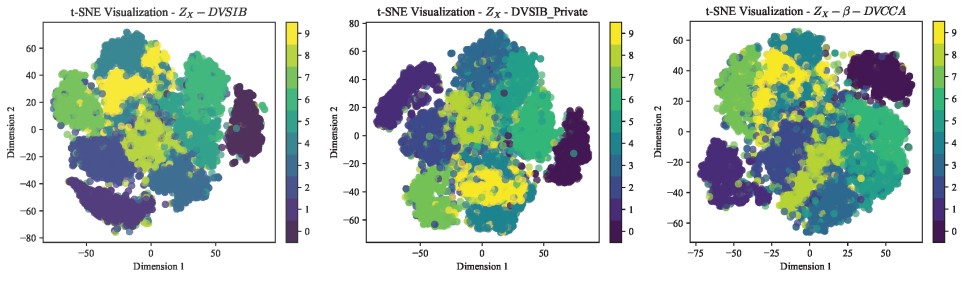

Figure 15: t-SNE X

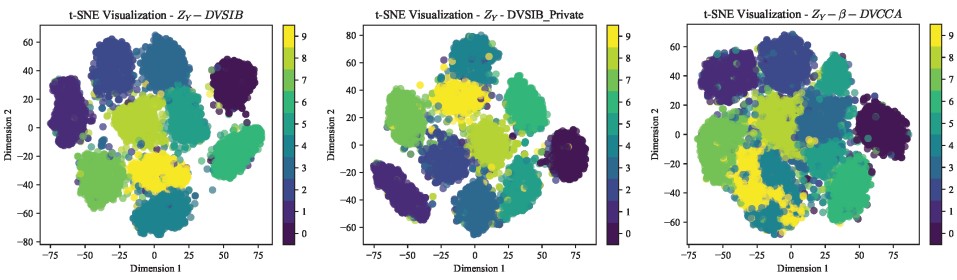

Figure 16: t-SNE Y

## E.3 DVSIB-private reconstructions for best parameters

Figure 17 shows the t-SNE embeddings of the private latent variables constructed by DVSIB-private, colored by the digit lable. To the extent that the labels do not cluster, private latent variables do not preserve the label information shared between $X$ and $Y$.

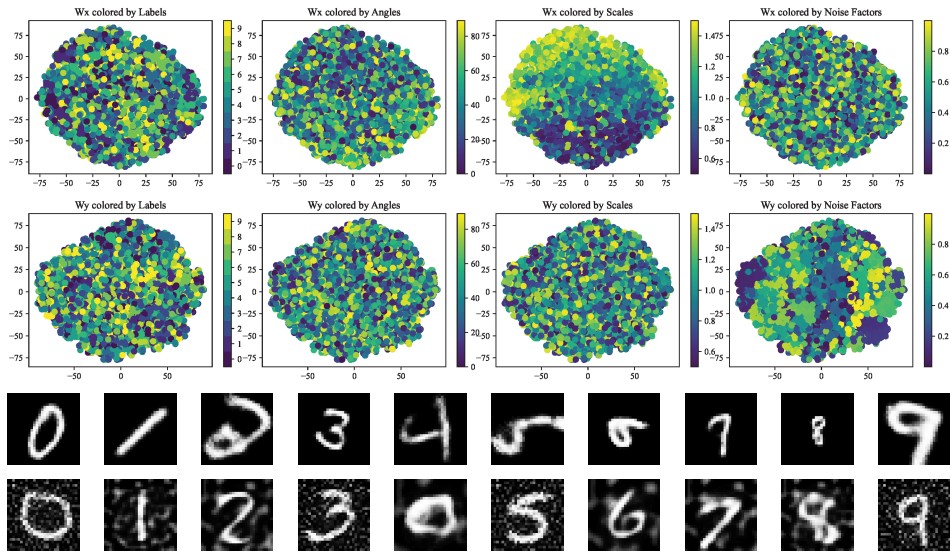

Figure 17: Private embeddings of DVSIB-private colored by labels, rotations, scales, and noise factors for $X$ (up), and $Y$ (middle). Reconstructions of the digits using *both* shared and private information (bottom) show that the private information allows to produce different backgrounds, scalings, and rotations.

## E.4 Additional results at 2 latent dimensions

We now demonstrate how different DR methods behave when the compressed variables are restricted to have not more than 2 dimensions, cf. Figs. 18, 19.

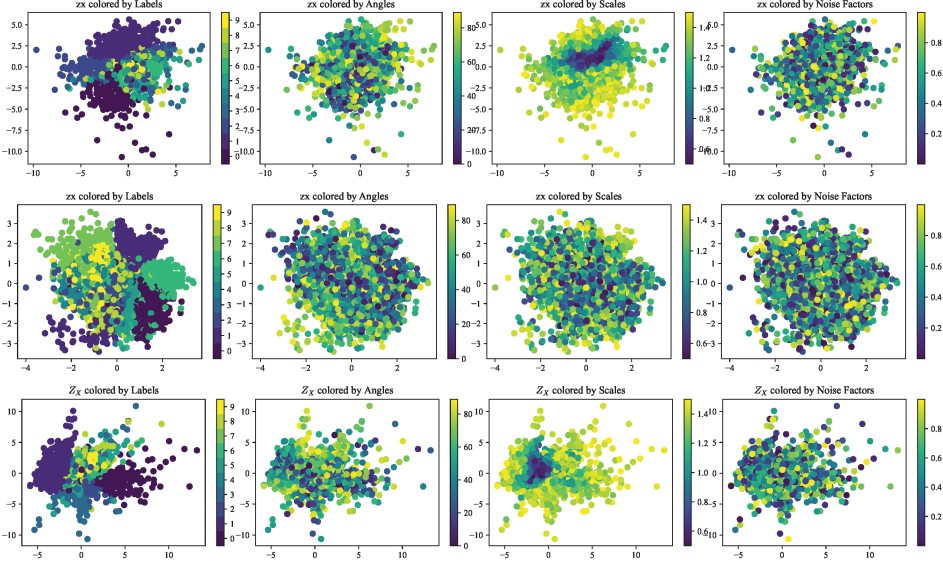

Figure 18: Clustering of embeddings when restricting $k_{Z_X}$ to two for DVSIB, DVSIB-private, and $\beta$-DVCCA, results on the $X$ dataset.

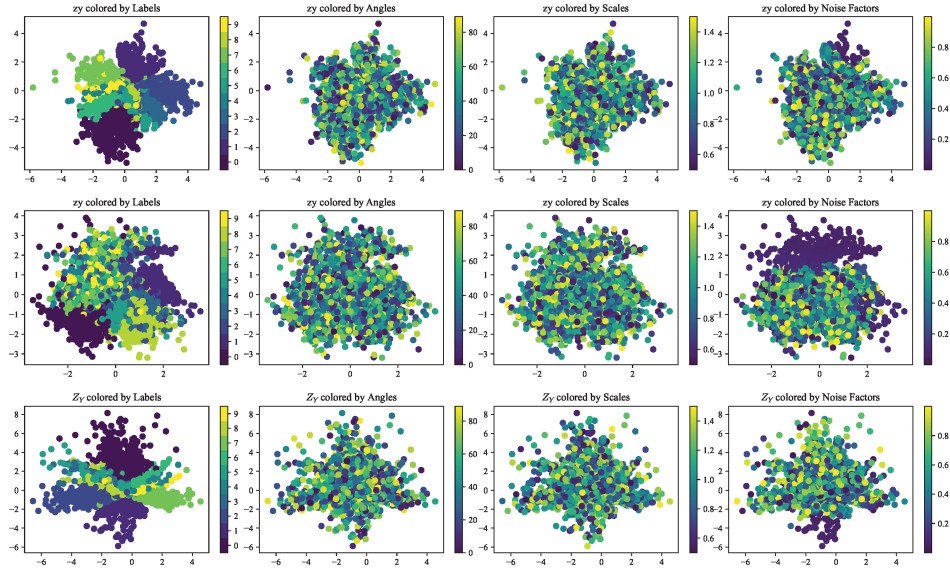

Figure 19: Clustering of embeddings when restricting $k_{Z_Y}$ to two for DVSIB, DVSIB-private, and $\beta$-DVCCA, results on the $Y$ dataset.

### E.5 DVSIB-PRIVATE RECONSTRUCTIONS AT 2 LATENT DIMENSIONS

Figure 20 shows the reconstructions of the private latent variables constructed by DVSIB-private, colored by the digit label, rotations, scales, and noise factors for $X$ (up), and $Y$ (bottom). Private latent variables at 2 latent dimensions preserve a little about the label information shared between $X$ and $Y$, but clearly preserve the scale information for $X$, even at only 2 latent dimensions.

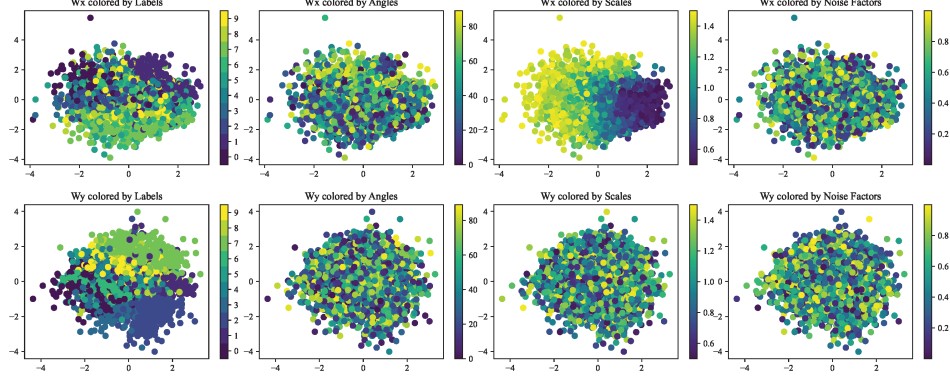

Figure 20: Private embeddings of DVSIB-private colored by labels, rotations, scales, and noise factors for $X$ (top), and $Y$ (bottom).

