# OpenReview forum: "Deep Variational Multivariate Information Bottleneck - A Framework for Variational Losses"
_ICLR.cc/2024/Conference — Submitted to ICLR 2024_

### Official Review · Reviewer_weLs · 2023-10-30

**Soundness:** 2 fair
**Presentation:** 2 fair
**Contribution:** 2 fair
**Rating:** 5
**Confidence:** 3

**Summary:**

The paper introduces a unifying principle for viewing and designing variational bounds based on multiinformation, aka total correlation [1]. The paper combines different variational upper and lower bounds (including MI Neural Estimators – MINE). Their frameworks recovers for example beta-VAEs or DVCCA. The newly introduced deep variational symmetric information bottleneck DVSIB objective is illustrated on a noisy MNIST dataset and where it yields to clusters for t-SNE embeddings or good classification accuracies based on the latent representations.


[1] Watanabe, Satosi. "Information theoretical analysis of multivariate correlation." IBM Journal of research and development 4.1 (1960): 66-82.

**Strengths:**

A plethora of works have been developed for multi-modal datasets that rely on variational methods in varying forms. Having a unifying framework to analyse such works is thus of great importance for the community. The submission thus addresses an important issue. Their introduced DVSIB objective is new, as far as I am aware. It outperforms other multi-modal variational methods for classifying noisy MNIST based on the latent representations.

**Weaknesses:**

The paper is sometimes difficult to follow and I feel that the structure of the paper can be improved, for example by using Definition/Proposition/Theorem etc.

To better assess the performance of DVSIB, it would be very useful to (i) compare it against previous work that also rely on multimodal information-theoretical measures,  such as [1] using a multiinformation bottleneck, (ii) evaluate not just whether the latents can be used for classification, but also the quality and cross-model consistency of the reconstructed images, for example following standard multi-modal evaluation measures [2]; and (iii) consider additional multi-modal datasets beyond noisy MNIST.

[1] Hwang, HyeongJoo, et al. "Multi-view representation learning via total correlation objective." Advances in Neural Information Processing Systems 34 (2021): 12194-12207.
[2] Shi, Yuge, Brooks Paige, and Philip Torr. "Variational mixture-of-experts autoencoders for multi-modal deep generative models." Advances in neural information processing systems32 (2019).

**Questions:**

Can the approach be generalised to more than two modalities?

How are the $\beta$ values in Table 2 tuned? Does it make sense to use different $\beta$ values for evaluating the classification accuracy for different methods, as I would guess that different $\beta$s impact how much information is encoded into the latent variables.

---

> ### Author Response · Authors · 2023-11-22
>
> We sincerely appreciate the time and effort you dedicated to evaluating our manuscript and for your thoughtful insights.
>
> $\\textbf{Structural concerns:}$ We acknowledge your suggestion about potentially framing the paper using Definition/Proposition/Theorem structures. However, we believe that our current outline, which delves into explaining the framework's mathematics, its implementation, the generation of new methods, and benchmarking, aims for pedagogical clarity. It seems that there's a significant disagreement among the Reviewers whether the structure of the manuscript that we've chosen is easy to understand, and we suspect that this is due to the Reviewers coming from different subfields. As we explained above, we chose to keep the current structure, aimed largely at those working in the Information Bottleneck space, and we also refined our manuscript to provide a more explicit explanation for readers from other backgrounds.
>
> $\\textbf{Comparison to other methods:}$ We appreciate your recommendation to compare against various multimodal architectures and evaluate cross-modal consistency. We had already included multimodal architectures such as CCA and its variants even in the original submission. Furthermore, we now integrated the architectures cited in your suggestions into our methods. As we have stated in the general response, it is important to note that our focus lies in the framework itself, enabling the integration and derivation of various methods. We do not aim for state-of-the-art classification accuracy against much better tuned and bigger AI systems. Yet, even with these caveats, our findings already indicate improved accuracy when the method's structure aligns with the expected statistical dependencies in the dataset.
>
> $\\textbf{Evaluation on other datasets:}$ Expanding our method's evaluation to other datasets, indeed, could strengthen our work. To address this, we applied DVSIB to the MIR-Flick dataset (please the the general response). However, the caveats remain that it is hard to benchmark a new, developing method, against state-of-the-art systems, fine-tuned using computational resources not generally available to us.
>
> $\\textbf{Including more than two modalities:}$ Our approach can, indeed, extend to include multiple modalities. We have updated the appendix to include implementations of such architectures, demonstrating the straightforward derivation of loss functions for multimodal architectures mentioned in the literature involving more than two modalities. And yet we can still use the elements of the code that we used in the other methods as well, ensuring a straightforward generalization towards crafting new methods that address specific research questions.
>
> $\\textbf{Tuning of $\\beta$:}$  We perform a sweep across the specified range of $\\beta$, selecting the value that corresponds to the highest classification accuracy. We document these sweeps comprehensively in the supplementary data (in the provided Methods Results.ipynb notebook). The varying $\\beta$ values indeed impact the information content differently for each model, necessitating tailored beta values for maximizing their potential and ensuring a fair comparison between different methods.
>
> We genuinely thank you once again for your invaluable feedback and for your insightful comments. Your suggestions have been instrumental in refining our work and enhancing its clarity.

---

### Official Review · Reviewer_uWuS · 2023-10-31

**Soundness:** 2 fair
**Presentation:** 4 excellent
**Contribution:** 2 fair
**Rating:** 6
**Confidence:** 3

**Summary:**

This paper provides a unifying framework for a number of variational dimensionality reduction techniques through the unifying lens of the information bottleneck.  They use this framework to quickly rederive several existing techniques, including a very slight generalization of one technique (Deep Variational Canonical Correlation Analysis).  They also derive an approach similar to DVCCA which they call Deep Variational Symmetric Information Bottleneck (DVSIB), which involves performing dimensionality reduction on two views of a set of sample simultaneously.  The novelty is that while trying to maximally compress the latent representation of each view (i.e., minimize the amount of information between the views and their latent representations) DVSIB also tries to maximize the information between the two latent representations.  They apply various methods to a variation on MNIST where each observation consists of two "views" of the same digit.  One view is a random sample from MNIST of that digit that is then randomly rotated.  The other view is another random draw of the same digit that is then randomly noised.  The authors find that classifiers trained on top of DVSIB representations outperform classifiers trained on top of different representations.

**Strengths:**

* The presentation is extremely clear.
* The unifying framework is a nice, conceptually clean way to unify a number of methods, and makes it easy to quickly derive loss functions for a fairly general family of dimensionality reduction techniques.
* DVSIB seems like a sensible and promising approach for finding probabilistic embeddings of multi-view data.
* Table 1 is a nice compendium of methods and concisely explains how these methods fit into the proposed framework.

**Weaknesses:**

Major:

* The evaluations and benchmarking felt limited, for a few reasons that I will detail in the next few comments.  First, the MNIST example feels somewhat contrived, and MNIST in general has become something of a toy dataset.  A more complex, more realistic application dataset would make the applicability of DVSIB more clear.
* Even on the MNIST application, I felt that the benchmarking was insufficient.  In particular, in Figure 3 and Table 2 it seems like there is almost no penalty for making $\beta$ huge as long as it is large enough, in which case the penalty on the encoder essentially does not matter.  In the VAE setup this would cause the encoder to concentrate on the MAP instead of the posterior, essentially reverting to an Auto-Encoder.  Are only the means of the variational distributions used in the downstream classification task?  Can the authors performing any benchmarking where the probabilistic interpretation on the latent space matters?
* Similarly, it seems like the MNIST benchmarking only gets at the importance of the size of the latent space in an oblique way.  In particular, across almost all methods performance always does better with a larger latent space (including DVSIB).  The paper would benefit from an analysis similar to the motivation suggested in the introduction, namely an example where the original dimensionality of the data is prohibitively large relative to the number of labeled examples for training the classifier, but a large unlabeled dataset exists to learn a good dimensionality reduction. In such a case dimensionality reduction would be absolutely necessary to obtain good classification performance.
* In order to compute the mutual information between $Z_X$ and $Z_Y$, the authors essentially use an energy-based model, learning $T(z_x, z_y)$ as an unnormalized log-likelihood (up to multiplication by the marginals).  Additional details on how the normalizing constant, $Z_\text{norm}$, is computed or approximated are warranted.

Minor / typos:

* Is $\Sigma_{Z_X}(x)$ assumed to be diagonal?  If not, what does it mean to learn the ``log variance''?
* I believe that the title of Subsection 2.2 should be "Variational Bounds" not "Variation Bounds"
* There is a typo in Equation (13) -- the MINE subscript should be on the term involving $Z_X$ and $Z_Y$, not the term with $Y$ and $Z_Y$.
* "available in the Appendix 5,4" appears to be a typo.

**Questions:**

see weaknesses

---

> ### Author Response · Authors · 2023-11-22
>
> We thank the Reviewer for the kind words regarding our presentation, framework, and method. Your recognition of the clarity in our presentation and the conceptual soundness of our unifying framework is immensely gratifying. We endeavored to present our work in a clear and coherent manner. We agree that DVSIB provides "a sensible and promising approach for finding probabilistic embeddings of multi-view data". This resonates deeply with our intentions to apply DVSIB to neural and behavioral data, which we are currently exploring in other research avenues.
>
> $\\textbf{Limited experimental evaluation:}$ Addressing the weaknesses you pointed out, particularly the limited evaluation, we want to reiterate that the core of our contribution lies in the framework itself. We aim to demonstrate how this framework provides unification, generalization, and how it facilitates the creation of new methods. We consider Table 1, as much a  primary result of our work as Table 2, which outlines how different algorithms perform on the MNIST data. We have never aimed to achieve state-of-the-art classification accuracy against much bigger and more fine-tuned learning systems. Tishby (one of the inventors of the now so popular Information Bottleneck framework) famously noted that the "tyranny of percent correct" limits the adoption of conceptual advances in machine learning since new, developing concepts cannot immediately compete with well-tuned older methods. We hope that the Reviewers will share this sentiment with us. Nevertheless, we are now including benchmarking of our DVSIB against well-tuned other methods on the MIR-Flicker dataset (see general response for details).
>
> $\\textbf{Choice of $\beta$:}$ The reviewer questioned the lack of performance penalty for large $\beta$ values. Our current understanding of this phenomenon is as follows. The training drives the decoder to maximize the information gathered from the obtained samples, influencing the reconstruction process significantly. Specifically, our approach involves starting with a batch of pairs of data points, feeding them into the encoder graph, and obtaining probability distributions of $p(z_X|x)$ and $p(z_Y|y)$. Subsequently, we sample from these distributions to learn the decoder. The parameter $\beta$ plays a crucial role in the decoder's behavior. A larger $\beta$ prompts the decoder to grasp as much information as possible from the samples $z_X$ and $z_Y$, pushing for maximal information retention in the reconstruction process, given the variational model and the dimensionality of the latent representations. Since the decoder has $I(Z_X, Z_Y)$ term, even at very large $\beta$, and with variational and dimensional constraints, the methods, therefore, $\\textit{does not}$ revert to an autoencoder since not all of the features of the original samples of views are useable for predicting the representation of the other view.
>
> $\\textbf{Size of the latent space:}$  The rationale behind increasing dimensions, indeed, allows capturing of finer-grained information between the two latent representations, which might not be easy to capture in the initial dimensions. This is especially true when the sample size is large enough, so that these small details stand out from the statistical noise (which is the case for our tests). However, in realistic scenarios with limited samples, expanding dimensions can hurt the reduction process. See for example (https://arxiv.org/abs/2309.05649) for the analysis of this problem (we expect that more dimensions will hurt, similar to how larger cardinalities do). In such cases, maintaining low dimensionality becomes pivotal for achieving both high accuracy and feasible modeling.
>
> $\\textbf{Estimating mutual information in the latent space:}$ Many estimators can be used to compute the information between $Z_X$ and $Z_Y$. The only requirement is that the estimator is differentiable. However, as we discussed above, it is important to use an estimator that "matches" the specific task at hand. As we hinted above, we leave the detailed analysis of which estimators should be used when to future work. Finally, in the revision we provided more explicit details regarding the computation of the normalizing constant, $\\mathcal{Z}_\\text{norm}$, as the Reviewer has requested.
>
> We would like to end with our appreciation of your meticulous detail in noting minor typos. We rectified these errors in our revised manuscript. Thank you for your detailed feedback.

---

### Official Review · Reviewer_JTcS · 2023-11-01

**Soundness:** 3 good
**Presentation:** 2 fair
**Contribution:** 2 fair
**Rating:** 5
**Confidence:** 5

**Summary:**

The paper introduces Deep Variational Multivariate Information Bottleneck (DVIB) as a framework to derive variational losses for dimensionality reduction purposes. A method section rooted in existing literature demonstrates how to de-compose multi-information associated with encoding and decoding distribution and how to bound and estimate each term using variational inference and deriving a Deep Variational Symmetric Information Bottleneck (DVSIB) objective for a specific instance of graphical models. The effectiveness of the proposed method and model is demonstrated on augmented pairs of MNIST digits.

**Strengths:**

1) The paper introduces a general framework inspired by the Information Bottleneck principle that can in theory applied to a wide variety of graphical models as an effective dimensionality (and information) reduction strategy.

2) The DVIB framework generalizes a variety of models in the literature, extending VIB [1] to graphical models with more than 2 variables.

**Weaknesses:**

## Main concerns

1) **Novelty**
    1) The novelty of the proposed Deep Variational Symmetric Information Bottleneck seems quite limited since the objective is quite similar to the existing literature and the main differences are not clearly underlined in the main text.

2) **Experimental analysis**
   1) The paper introduces a framework that can in principle applied to complex graphical models involving multiple variables, but the experimental section (and most of the method) solely focuses on a two-variable system that has been widely explored in the literature.
   2) The experiments revolve solely around the MNIST dataset. Further, the paper claims that "none of the algorithms were given the data labels" even though the training pairs are constructed by pairing digits with the same label. As a result, label information is indirectly captured in the dataset structure.
   3) The paper lacks common baselines based on contrastive learning that can be applied in the same settings [1,2,3]. In particular [2] proposes a similar loss function and demonstrates similar performance without using the labels for pairing images.
   4) The qualitative visualization relies solely on t-SNE even if there is evidence to support that t-SNE visualization could be misleading [4].

The paper presents an interesting approach through the DVIB framework, which holds the potential for principled dimensionality reduction in structured datasets consisting of tuples of joint observations. However, the current submission falls short of demonstrating its contributions due to a limited experimental section and lack of novelty in the chosen setting. A more compelling case could be made by extending the analysis and experiments to encompass more complex graphical models and tasks, as opposed to the limited scope of addressing well-studied symmetric 2-observed variables as reported in the main text. This expanded focus would not only enhance the novelty but also demonstrate the method's applicability and effectiveness in more challenging scenarios.

## Minor issues
1) Some of the citation years and venues are incorrect (e.g. Friedman et al., 2013 has been published in UAI 2001)


### References
[1] Chen, Ting, et al. "A simple framework for contrastive learning of visual representations." International conference on machine learning. PMLR, 2020.

[2] Federici, M., Dutta, A., Forré, P., Kushman, N., & Akata, Z. "Learning robust representations via multi-view information bottleneck." International Conference on Learning Representations, ICLR, 2020.

[3] Zbontar, Jure, et al. "Barlow twins: Self-supervised learning via redundancy reduction." International Conference on Machine Learning. PMLR, 2021.

[4] Yang, Zhirong, Yuwei Chen, and Jukka Corander. "T-SNE is not optimized to reveal clusters in data." arXiv preprint arXiv:2110.02573 (2021).

**Questions:**

1) What are the main differences between DVSIB and the existing methods in literature? How are they potentially related to the improved performance?

2) What is the rationale behind the choice of MINE for mutual information maximization? More recent mutual information maximization strategies [1] are shown to yield more stable and effective training.

3) How does the prescribed DVIB model perform on more complex datasets consisting of tuples of observations with a known graphical model? Can DVIB make better use of the structure of the problem when compared to popular modern representation learning methods that do not explicitly consider the relation between the variables?


### References

[1] Poole, Ben, et al. "On variational bounds of mutual information." International Conference on Machine Learning. PMLR, 2019.

---

> ### Author Response · Authors · 2023-11-22
>
> We thank the Reviewer for their careful assessment of our work.
>
> $\\textbf{Novelty:}$ We believe that an important contribution of our work is in providing a conceptually clean way to unify a number of dimensionality reduction methods within the information-theoretic language. For example, trade-off parameters that often need to be added in an $\\textit{ad-hoc}$ manner with other derivations naturally arise using our framework. As we emphasized in our previous responses, the Deep Variational Symmetric Bottleneck is meant to be one of the applications of our more general framework. It produces two latent variables that retain maximal information about one another, rather than about the the individual views of the data. This is a main distinction of this specific method from the others we reviewed.  Part of the loss of DVSIB anchors the $Z_X$ and $Z_Y$ via decoders to $X$ and $Y$. This is one main difference as compared to your Ref.[2]. Our method produces a generative model for $X$ and $Y$ at the same time as producing $Z_X$ and $Z_Y$ maximally informative of each other. Ref.[2], on the other hand, makes distributions for mapping of the latent variables $Z_X$ and $Z_Y$ as close to one another as possible. In fact, it enforces that $Z_X$ and $Z_Y$ have the same domain. This is different from DVSIB where the latent variables can have different units, dimensions, and domains. Please see our general response for details. Finally, we now include performance of the loss in Ref.[2]  loss in our main results table for comparison, and we thank the Reviewer for the suggestion that we should compare to this and other additional methods.
>
> $\\textbf{Additional evaluation:}$ We now evaluate our method using the MIR-FLICKR dataset. Please see the general response for the details of what we did. Further, we would like to clarify that the noisy MNIST dataset that we used for our main experiments was not given the data labels. The label information, of course, is indirectly captured in the dataset structure by having pairs of images with the same label. This is precisely the main point of DVSIB and related multiview methods:  if there is some relationship between $X$ and $Y$, we can recover it without telling our model about the relationship (aka, without labeling). We were able to demonstrate that DVSIB  was able to do just that by elevating the performance of a linear SVM to predict the labels from the latent space. As far as we understand, this is a standard way of testing representation learning algorithms (see e.g., Wang et al., 2015; 2016).
>
> $\\textbf{Concerns about visualization:}$ We recognize that t-SNE is not infallible for showing evidence of clustering or of a quality of the latent space. Hence we use t-SNE for qualitative visualization only, and our main results are quantified instead by training an SVM on our learned latent space for label identification. Our SI includes several plots of visualizations of our method reduced to two dimensions where t-SNE is not used, yielding qualitatively similar results.
>
> $\\textbf{Question 1:}$ We answered this question in more detail in our main response to all Reviewers.
>
> $\\textbf{Question 2:}$ MINE is one possible estimator that we could have used to produce a variational bound on the information between the latent representations. $I_\\text{NCE}$ could just as easily have taken its place (parenthetically, existing work shows that the two are related, for instance [3]). Our approach just requires a differentiable estimator. Neither of the methods will be good universally, and different estimators should be used in different applications, depending on properties of the data. We now note this in the paper text. However, for the demonstrations in this work, MINE was sufficient, and we decided to stay with it.
>
> $\\textbf{Question 3:}$ We believe this is an important question, and one that we are addressing in future work. We think that providing the dependence structure of the problem (as we do using the encoder and the decoder graphs) that matches the data in question will yield better methods, more tailored to their respective problems. However, none of the methods will be better than the others universally. We are working on applications in the context of neuroscience and dynamical systems inference, where we will explore this question in detail in specific datasets.

---

### Official Review · Reviewer_MfdL · 2023-11-05

**Soundness:** 2 fair
**Presentation:** 2 fair
**Contribution:** 3 good
**Rating:** 5
**Confidence:** 3

**Summary:**

The authors study variational dimensionality reduction and propose a multivariate information bottleneck framework that generalizes several existing method (e.g, beta-VAE) and yields new algorithms for settings where we want to jointly compress two distinct data representations.

**Strengths:**

- The authors study an important problem in dimensionality reudction
- The proposed framework yields a nice generalization of existing methods for variational dimensionality reduction

**Weaknesses:**

1. I feel like the writing of the paper still has quite a bit of room for improvement
- Even understanding the task that the authors are solving took a long time. I see it first in Section 2.1. It should be clear from the abstract and intro that we are trying to map two views into a latent space. Right now, the intro reads like a long list of related work.
- I am familiar with the VAE and variational inference literature, and I found the derivations unnecessarily hard to follow. In particular, there exists standard notation used in variational inference (e.g., q is an encoder/approximate posterior, p is the decoder, etc.) and it doesn't seem to be used in the paper.
- The paper is trying to solve problems that are in the domain of probabilistic modeling and variational inference, but uses techniques based on information theory. For readers that are less familiar with information theory, it would help to have a paragraph that explains more how these methods relate to the literature on VAEs and variational infernece.
- At a high level, I found that in some places the paper is overly verbose, and in others it is overly terse.

2. The experimental results are not very strong in my opinion.
- First of all, since this is not mainly a theory paper, I feel like the authors should experiment on more than one dataset.
- Ideally, some of these datasets would be more sophisticated than MNIST. I feel like this method would be very useful for researchers in biology or neuroscience, perhaps exploring applied problems in these fields would make the paper stronger.
- I am not entirely sure if the set of baselines is the best one. For example, the new method is the only one which defines two separate latents for each of the two views of the data. Is the improvement in performance attributed to the fact that each view gets its own latent (which is not a novel idea from this paper; there are other methods that do this), or to the specific way in which the method generates these latents. In order to determine this, another baseline that computes one latent variable per view would be helpful.
- In particular, what if I were to fit a VAE-type model with two latents $Z_X, Z_Y$ and two observed variables $X, Y$ such that the q and the p have the same independence structure as DVSIB in Table 1. Would this approach be equivalent to DVSIB? If yes, there should be a discussion. If they are not equivalent, then the VAE-type model should be a baseline (and there should still be a discussion of the pros/cons of each approach).

Overall, I am leaning towards rejection, but I am willing to raise my score if I am missing something, or if there is additional compelling evidence that the authors could provide.

**Questions:**

- Are there any additional datasets and baselines that could be added to the paper?
- How does the method compare to VAE-type model with two latents $Z_X, Z_Y$ and two observed variables $X, Y$ such that the q and the p have the same independence structure as DVSIB in Table 1?

---

> ### Author Response · Authors · 2023-11-22
>
> $\\textbf{Presentation:}$ We thank the Reviewer for their suggestions for improving the paper. Existing literature for autoencoders and VAEs typically uses maximum likelihood methods for deriving the loss functions. Here we use an information bottleneck approach to achieve the same thing. The Information in the Decoder graph is maximized, which corresponds to maximizing the log-likelihood, and the information in the encoder graph is minimized corresponding to the approximate posterior.  We follow the notation used by Alemi et al. in "Deep Variational Information Bottleneck", which has become a $\\textit{de facto}$ standard in the context of information-based variational models.  Thus we use $p$ to represent the "true" distributions and $q$ and $r$ as variational approximations to the true distributions. We understand that this is not the notation that is used commonly in the VAE literature, which makes our paper difficult to read for some members of the community.  However, for some other members, e.g., Reviewer 4, this choice seems to have made the presentation clear. We have made the choice to stay with the information-based notation. However, to make our work accessible to the other half of the community, we added additional details in the main text describing our notation and how our framework relates to the literature on VAEs. We hope that this will make the paper easier to read for everyone.
>
> $\\textbf{Impact and contribution:}$ One of our primary contributions is the proposal for the general Deep Variational Multivariate Information Bottleneck (DVMIB) framework. This is a general framework, which incorporates many of the representation learning methods (the ones we explored in the paper, and most of the ones mentioned by the Reviewers) under one information-theoretic umbrella, allowing to design new methods systematically instead of $\textit{ad hoc}$. In fact our second primary contribution was in using this flexibility of the framework to create a new specific method, DVSIB, which has not been seen anywhere in the prior literature. DVSIB is just an example of what the framework can do, and, while novel and promising, we only view it as the second main contribution, behind developing a conceptually clean and flexible framework for designing variational losses. We have clarified our main contributions in the introduction to our paper.
>
> $\\textbf{Additional experiments:}$ We agree with the Reviewer that evaluation of DVSIB on more complex datasets is welcome. Crucially,  we want to reiterate that we do not intend to compete with the state-of-the-art classification methods, trained using much bigger networks, on hardware beyond those available to us, and hence with the ability to fine-tune all algorithmic parameters. As we explain in our general response, we provide the current implementation of DVSIB as a proof of concept, hoping to illustrate that, even without fine-tuning and large-scale computing, its performance is already promising. To this end, we now evaluate DVSIB on an additional, more complex dataset: the MIR-FLICKR dataset. Please see the general response for more details of what we've done. We also conducted additional experiments on other algorithms suggested by this Reviewer and other Reviewers. The main Table in the manuscript already included several methods that computed one or more latent variable per view. including DVCCA and its variants. The table now includes performance using the loss from "[1] Learning robust representations via multi-view information bottleneck" paper, among others suggested by the Reviewers.
>
> $\\textbf{More on relations to VAEs:}$ With regards to your question, "If I were to fit a VAE-type model with two latents $Z_X$, $Z_Y$ and two observed variables $X$, $Y$ such that the $q$ and the $p$ have the same independence structure as DVSIB'', we believe that this would be the same as our approach, with just a different name for the same concept. (This is as long as there is the same dependence between $Z_X$ and $Z_Y$ as shown in our model of DVSIB. If there is no dependence then, the structure you described is the same as two separate VAEs, one for $X$ and one for $Y$). This joint encoding, while keeping information between the compressed variables is crucially different from parallel VAEs in our opinion, as we have emphasized in the general response.

---

### Author Response · Authors · 2023-11-22
**Response to all Reviewers**

$\textbf{Part 1/2}$

We genuinely thank all the Reviewers for their detailed and careful responses. Your insights and suggestions have been instrumental in improving the quality of our paper, and we deeply appreciate the time and effort you've dedicated to reviewing our paper.

We start this response by first commenting on critiques common to multiple reviews, and then we focus on individual responses.

$\textbf{Diversity of methods:}$ Multiple Reviewers pointed at many additional related methods, underscoring the vast diversity of methods within the field. Exploring every possible pairing of relationships between data modalities for compression or reconstruction presents a complex landscape. And there's a clear need for frameworks that can systematize all (or most) of these methods. As we argue, our approach does just that --- and this is one of the main significance of our work. In the response to Reviewes, we discuss every one of the methods mentioned by the Reviewers within the context of our framework, and point out similarities and differences of these approaches.

$\textbf{More complex multi-view models:}$ We aim to unite various methods mentioned by the Reviewers within one framework. While we focused on scenarios with two modalities, our framework accommodates large number of views as well. In the revision, we added discussions of methods with more modalities within our framework, explaining also how the approach opens room for their straightforward improvement.

$\textbf{Impact of our paper:}$ We see multiple contributions to impact. First, our framework facilitates a straightforward connection among multiple dimensionality reduction methods. By graphically representing statistical dependencies in encoding and decoding schemes, our framework allows nearly automatic translation of these schemes into practical implementations. Second, our framework incorporates a tradeoff parameter between compression and fidelity of decoding, which allows for a natural generalization of many existing algorithms. Third, in DVSIB---a specific example of our more general framework---we focus on distinct latent representations of both data views. We believe that having such distinct latent spaces is essential in interpretation of latent representations, especially in sciences, where each of the views and their latent embeddings may then be expressed in different units, independently measurable in experiments. While there are other algorithms that seek independent embeddings of views, there these embeddings are trained independently. It was recently shown that this is not data-efficient, (https://arxiv.org/abs/2309.05649).  DVSIB dimensionality reduction is simultaneous, thus requiring less data, and allowing to extract statistical dependencies between the latent representation of different views directly during training. This holds promise in various scientific inquiries, particularly, as Reviewers pointed out, in fields like neuroscience and dynamical inference.

$\textbf{Incorporating revisions:}$ We have taken note of both minor and major concerns raised by the Reviewers, and they are all addressed in the revised manuscript.

$\textbf{Application of methods to noisy MNIST:}$ While testing new and previously developed methods against this simple dataset, it is important to clarify our intent. Our aim is not to achieve state-of-the-art classification or compression. Instead, our primary focus is on testing different representation learning methods on the same well-understood dataset, and with neural networks and optimization algorithms of the same complexity for all methods, so that we are comparing the methods, in general, rather than their specific detailed representations.

---

> ### Author Response · Authors · 2023-11-22
>
> $\textbf{Part 2/2}$
>
> $\textbf{Broader testing - Application to MIR-Flickr dataset:}$ As an additional---and a more complex---test case for different DR methods within our framework, in the revised manuscript, we now introduce a new appendix, which tests DVSIB on the MIR-Flickr dataset.
>
> The MIR-Flickr dataset consists of one million images and 800,000 tags. Each image consists of 3857 hand-crafted features and the 2000 most frequent tags are encoded into a 2000-dimensional binary vector indicating presence or absence of the tag for an image. The dataset is further divided into 25,000 images that are annotated as belonging to 38 possible topics along with their tags. The remaining 975K image-tag pairs are considered unlabelled. On average, an image belongs to 4.7 topic categories. Following previous literature, we remove images that have fewer than two tags resulting in about 750 thousand image-tag pairs.
>
> We evaluate DVSIB on the MIR-Flickr dataset using a standard procedure outlined in the literature. We train DVSIB on the unlabeled pairs of images and tags to produce two latent spaces, one for images and the other for tags. We then use the trained encoders to encode the labeled set of images, tags, and topics into the latent spaces trained on unlabeled data. The trained encoders output two 64 dimensional latent spaces (one for the image and one for the tags). We then verify how meaningful the representations learned on unlabeled data are for the labeled data. For this, we subdivide the set of 25,000 labeled images into train, test, and validation sets of 10K, 5K, and 10K images, respectively. We train and test a multi-label logistic classifier from the latent representations of the data to the topics. We then evaluate the mean precision of the classifiers based on the latent image space and the latent tag space, both learned from unlabeled data, on the labeled validation data subset. The image latent space has a mean precision of 0.682 and the tag latent space has a mean precision of 0.712. This compares to a reported precision of 0.529 CCA, 0.565 Contrastive, 0.573 DCCA, 0.751 for MV-InfoMax, and 0.749 for the multi-view information bottleneck [1]. Note that the precision for the latter two methods is higher than for DVSIB. However, this was achieved with much larger dimensionality of the latent spaces (1024 to 64 in DVSIB), fine-tuning the trade-off parameter $\beta$, deeper networks for variational approximations (4 hidden layers to 1 in our work), as well as fine-tuning many parameters of the optimization algorithms. In contrast, in the short time we had for this revision, we had no opportunity to fine-tune DVSIB. We believe that the performance of the un-tuned DVSIB in the middle of the pack of other methods -all fine-tuned, and implemented with much larger networks- is remarkable, and bodes well for its future successes.
>
> $\textbf{To Conclude:}$ We believe that our work has a potential to unify many disparate approaches to DR seen in the literature, providing for uniform comparisons and for principled selection of methods to use in specific scenarios. The developed framework will allow proposing and implementing new DR methods, not yet known. It also enables methods for simultaneous inference of distinct representations of different data views, and for finding statistical dependencies between the compressions directly, which has not been possible previously.

---

### Author Response · Authors · 2023-11-22
**Papers Mentioned in Reviews**

$\\textbf{Part 1/3}$

In the following comment we briefly review the literature that was mentioned in the Reviews. We compare the methods to our general framework, and to  DVSIB specifically.

$\textbf{ [1] Federici et al., "Learning robust representations via multi-view information bottleneck." ICLR, 2020}$

The paper defines a multiview information bottleneck (MVIB) by minimizing any redundant information between views $(v_1, v_2)$. This is achieved with the following losses:
\\begin{align}
L_1 &= I(z_1;v_1|v_2)-\\lambda_1 I(v_2; z_1),\\\\
L_2 &= I(z_2;v_2|v_1)-\\lambda_1 I(v_1; z_2).
\\end{align}
These losses are equivalent to two deep variational information bottlenecks (as in (https://arxiv.org/abs/1612.00410)), performed in parallel. Within our framework, the same algorithm emerges with the encoder graph that compresses $v_1$ into $z_1$ and $v_2$ into $z_2$, while the decoder graph would reconstruct $v_2$ from $z_1$ and $v_1$ from $z_2$.

The authors then combine these two losses while enforcing the condition that $z_1$ and $z_2$ are the same, and they additionally bound the combined loss function to obtain:
\\begin{equation}
L_{\\rm MIB}=-I(z_1,z_2)+D_{SKL}(P(z_1|v_1)||P(z_2|v_2)),
\\end{equation}
(recall, again, that $z_1$ and $z_2$ should be the same latent space in this approximation). Here $D_{\\rm SKL}$ is the symmetrized KL divergence, $v_i$ correspond to the two different views, and $z_i$ correspond to their two latent, compressed representation. While this loss looks similar to the DVSIB loss, it is conceptually different. It attempts to produce latent variables that are as similar to one another as possible (ideally, $z_1=z_2$). In contrast, DVSIB attempts to produce different latent variables that could in theory have different units, dimensionalities, and domains, while still being as informative about each other as possible. For example, in the noisy MNIST, $Z_X$ contains information about the labels, the angles, and the scale of images (all needed for reconstructing $X$) and no information about the noise structure. At the same time, $Z_Y$ contains information about the labels and the noise factor only (both needed to reconstruct $Y$). See Appendix for 2-d latent spaces colored by these variables, illustrating the difference between $Z_X$ and $Z_Y$ in DVSIB.
Further, in practice, implementation of MVIB in the original paper uses the same encoder for both views of the data; this is equivalent to encoding different views using the same function and then trying to force the output to be as close as possible to each other, in difference to what we have in DVSIB, which uses two distinct encoders.

In the original paper, MVIB was tested on noisy MNIST, on the Sketchy dataset, and on the MIR-FLICKR dataset. Their method's performance is similar to ours on noisy MNIST even though they used much larger networks, larger latent spaces, performed a lot of fine-tuning, and implemented more sophisticated training. While MVIB performance on FLICKR is better than DVSIB, we attribute this difference solely to the size and fine-tuning of the model, as we explained above.

$\\textbf{[2] Zbontar, et al. "Barlow twins: Self-supervised learning via redundancy reduction." PMLR, 2021}$

The Barlow twin architecture consists of identical twin networks fed with different distorted views of the same object. The objective functions goal is to make the cross correlation matrix between the embeddings produced by the twin networks be as close to the identity matrix as possible. This is a form of redundancy reduction typically used for self-supervised learning. The authors point out in the appendix that this architecture is equivalent to the information bottleneck in certain regimes, deeming it conceptually similar to the variational information bottleneck, which we already derived and evaluated within our framework. One of the main benefits of the twins networks is that this method does not require large batches of data to be successfully trained. We do not evaluate the Barlow twin architecture in our work. However, we did evaluate the deep variational information bottleneck and showed the superiority of DVSIB over it for building representations from multi-view data.

---

> ### Author Response · Authors · 2023-11-22
>
> $\\textbf{Part 2/3}$
>
> $\textbf{[3] Poole et al. "On variational bounds of mutual information." PMLR, 2019}$
>
> This paper provides several new bounds for variational information estimation including $I_{\\rm NCE}$. It provides a justification for the gradient trick used in the original MINE paper, defining a new model that provides the same gradient and loss up to a constant.  Furthermore it compares multiple mutual information estimators $I_{\\rm NWJ}$, $I_{\\rm NCE}$, $I_\\alpha$, and $I_{\\rm JS}$. These estimators vary between high variance for $I_{\\rm NWJ}$ and high biases for $I_{\\rm NCE}$. In fact the highest mutual information that can be captured by many of these methods is $\\log(\\frac{\\rm batch}{\\alpha})$. We can use any of these methods to estimate $I(Z_X,Z_Y)$ in our framework. In fact, we are currently exploring advantages and limitations of these different methods for a subsequent publication. While we are not ready to issue the final verdict, it is clear that some of these more complicated methods do not work well for estimating information between latent variables. Thus, for the current paper, we chose to use MINE as it is simple to understand and implement, and we decided to leave the cross-method comparisons for the future.
>
> $\\textbf{[4] Hwang et al. "Multi-view representation learning via total correlation objective." Advances in Neural Information Processing Systems 34 (2021)}$
>
> This paper introduces a broad multi-view information loss function, somewhat similar to our approach. It aims to uncover a shared representation across different views. Specifically, the paper focuses on maximizing the total correlation objective function in a convex optimization problem (not the regular information bottleneck, although there is some resemblance):
> \\begin{align}
> TC(\\vec{O};Z)\\ge \\frac{V-\\alpha}{V}\\sum_{v=1}^V [ H(O_v) + E_{p(z|\\vec{O})}p_D(\\vec{O})][\\ln q(o_v|z)]] - \\frac{\\alpha}{V} E_{P_D(\\vec{})}[D_{KL}[p(z|O)||r(z|O)]]
> -(1-\\alpha)E_{P_D(\\vec{})}[D_{KL}[p(z|O)||r(z)]].
> \\end{align}
>
> This function balances between a conditional VIB and a VIB for each view. It looks at a difference between the information of the joint view $\\vec{O}$ and $Z$ ($I(\\vec{O};Z)$) and the sum of information of independent views $O_\\nu$ and $Z$ ($I(O_\\nu;Z)$). Conceptually, this is similar (but not equivalent) to an encoder mapping from independent views $O_\\nu$ to $Z$, and a decoder from $Z$ to $\\vec{O}$, or conversely, encoder from a multiview and decoder to individual views, analogous to the Joint-DVCCA discussed in our paper. Thus, conceptually, this approach can be represented within our framework. There are quite a few conceptually similar approaches, such as Sutter et al. ("Generalized Multimodal ELBO"), Lee and Van Der Schaar ("A variational information bottleneck approach to multi-omics data integration"), Wan et al. ("Multi-View Information-Bottleneck Representation Learning"), Huang et al. ("On the Multi-View Information Bottleneck Representation"). All of these methods explore connections among individual data views, their joint representations, individual latent spaces, and their amalgamation. As we have emphasized, our important contribution is providing a conceptually clear framework to systematize all of these approaches, and to clearly see differences between them in terms of the encoder and decoder dependecies graphs.
>
> Nonetheless, the similarities between our approach and this one are not perfect. For example, it is not clear how to encode multiple representations interacting with each other (as DVSIB cab) in the total correlations picture, while dealing with conditional informations is not trivial in our framework. Thus understanding similarities and differences between the methods requires additional analysis, which we would like to postpone to future work.

---

> > ### Author Response · Authors · 2023-11-22
> >
> > $\\textbf{Part 3/3}$
> >
> > $\\textbf{[5] Shi et al,. "Variational mixture-of-experts autoencoders for multi-modal deep generative models." Advances in neural information processing systems32 (2019)}$
> >
> > This paper proposes a multi-modal VAE to learn a generative model for multimodal data. The aim is to satisfy four criteria: latent factorization (latent space factorizes into private and shared), coherent joint generation (generation of different modal views of the same object from the same latent variable), coherent cross generation (the model can generate data in other modalities given data from one modality), synergy (more views produce better results on all downstream tasks). This work can be represented as a special case of our conceptual framework (as ref [4]), where encoders are trained on various combinations of the views, and the decoder reconstructs each view from the latent variable.
> >
> > $\textbf{[6] Yang et al,. "T-SNE is not optimized to reveal clusters in data." arXiv preprint arXiv:2110.02573 (2021)}$
> >
> > This paper was mentioned in the context of us relying on t-SNE for visualization. It presents several example datasets where the data can be clustered successfully into meaningful clusters, but where t-SNE does not produce visually salient clusters. They additionally check the assumptions for the clustering of t-SNE and find that these assumptions are often violated. The understanding that t-SNE is a tool that often produces misleading results and hence must be used with care has certainly permeated the literature, and we are aware of these concerns. We stress that we used t-SNE for simple visualization only, but all of our main results are quantified more objectively by using DVSIB (and other methods) to pretrain embeddings, and then using classification via logistic regression or a linear SVM trained on the latent variables to produce objective accuracy scores.
> >
> > $\textbf{[7] Chen et al. "A simple framework for contrastive learning of visual representations." PMLR, 2020.}$
> >
> > This paper explores contrastive learning on ImageNet. It works by maximizing agreement between different views of the same data, as well as penalizing pairs that are negative examples. This is a state-of-the-art model for self-supervised clustering tasks in ImageNet. In this paper, ImageNet is augmented by several transformations (random cropping, random color distortions, and random Gaussian blur). Overall, the goal of this work is distinct from ours (self-supervised clustering vs. building latent spaces). Further, our aim has never been to train a state-of-the-art system on model datasets, but to develop a conceptual framework for designing variational bottleneck representation learning, which can be converted into a practical code. Thus we do not aim to compare raw performance of, say, DVSIB with state-of-the-art, much larger learning systems like the one in this reference.

---

### Meta-Review · Area_Chair_DhGV · 2023-12-09

**Metareview:**

The paper presents a learning framework for deep generative models involving multiple (hidden) variables, using a information bottleneck objective. The authors show that with specific application scenario (unsupervised, supervised, unsupervised multi-view), the framework encompasses several previously proposed methods, with perhaps additional trade-off parameters (\beta). Overall the differences between proposed method from previous methods are not too large. As an example, in the multi-view unsupervised setup, the proposed method reduces to the DVSIB loss which is similar to DVCCA and Federici et al, 2020 (MVIB), with main difference being this work using MINE estimation between (Z_x, Z_y), versus other matching loss in the latent spaces. The reviewers think the paper needs more empirical study and more theoretical analysis to validate the differences, so as to demonstrate novel technical contributions.

**Justification For Why Not Higher Score:**

In the current form, the novelty is moderate and the reviewers agree on the paper needing more empirical study.

**Justification For Why Not Lower Score:**

N/A

---

### Decision · Program_Chairs · 2024-01-16

Reject